# On The Relative Error of Random Fourier Features for Preserving Kernel Distance

**Kuan Cheng**
Peking University
Email: `ckkcdh@pku.edu.cn`

**Shaofeng H.-C. Jiang**
Peking University
Email: `shaofeng.jiang@pku.edu.cn`

**Luojian Wei**
Peking University
Email: `luojianwei@pku.edu.cn`

**Zhide Wei**
Peking University
Email: `zhidewei@pku.edu.cn`

## Abstract

The method of random Fourier features (RFF), proposed in a seminal paper by Rahimi and Recht (NIPS'07), is a powerful technique to find approximate low-dimensional representations of points in (high-dimensional) kernel space, for shift-invariant kernels. While RFF has been analyzed under various notions of error guarantee, the ability to preserve the kernel distance with *relative* error is less understood. We show that for a significant range of kernels, including the well-known Laplacian kernels, RFF cannot approximate the kernel distance with small relative error using low dimensions. We complement this by showing as long as the shift-invariant kernel is analytic, RFF with $\mathrm{poly}(\varepsilon^{-1} \log n)$ dimensions achieves $\varepsilon$-relative error for pairwise kernel distance of $n$ points, and the dimension bound is improved to $\mathrm{poly}(\varepsilon^{-1} \log k)$ for the specific application of kernel $k$-means. Finally, going beyond RFF, we make the first step towards data-oblivious dimension-reduction for general shift-invariant kernels, and we obtain a similar $\mathrm{poly}(\varepsilon^{-1} \log n)$ dimension bound for Laplacian kernels. We also validate the dimension-error tradeoff of our methods on simulated datasets, and they demonstrate superior performance compared with other popular methods including random-projection and Nyström methods.

## 1 Introduction

We study the ability of the random Fourier features (RFF) method (Rahimi & Recht, 2007) for preserving the relative error for the *kernel* distance. Kernel method (Schölkopf & Smola, 2002) is a systematic way to map the input data into a (indefinitely) high dimensional feature space to introduce richer structures, such as non-linearity. In particular, for a set of $n$ data points $P$, a kernel function $K : P \times P \to \mathbb{R}$ implicitly defines a *feature mapping* $\varphi : P \to \mathcal{H}$ to a feature space $\mathcal{H}$ which is a Hilbert space, such that $\forall x, y, K(x, y) = \langle \varphi(x), \varphi(y) \rangle$. Kernel methods have been successfully applied to classical machine learning (Boser et al., 1992; Schölkopf et al., 1998; Girolami, 2002), and it has been recently established that in a certain sense the behavior of neural networks may be modeled as a kernel (Jacot et al., 2018).

Despite the superior power and wide applicability, the scalability has been an outstanding issue of applying kernel methods. Specifically, the representation of data points in the feature space is only implicit, and solving for the explicit representation, which is crucially required in many algorithms, takes at least $\Omega(n^2)$ time in the worst case. While for many problems such as kernel SVM, it is possible to apply the so-called "kernel trick" to rewrite the objective in terms of $K(x, y)$, the explicit representation is still often preferred, since the representation is compatible with a larger range of solvers/algorithms which allows better efficiency.

In a seminal work (Rahimi & Recht, 2007), Rahimi and Recht addressed this issue by introducing the method of random Fourier features (see Section 2 for a detailed description), to compute an explicit low-dimensional mapping $\varphi' : P \to \mathbb{R}^D$ (for $D \ll n$) such that $\langle \varphi'(x), \varphi'(y) \rangle \approx \langle \varphi(x), \varphi(y) \rangle =$

$K(x, y)$, for shift-invariant kernels (i.e., there exists $K : P \to \mathbb{R}$, such that $K(x, y) = K(x - y)$) which includes widely-used Gaussian kernels, Cauchy kernels and Laplacian kernels.

Towards understanding this fundamental method of RFF, a long line of research has focused on analyzing the tradeoff between the target dimension $D$ and the accuracy of approximating $K$ under certain error measures. This includes additive error $\max_{x,y} |\langle \varphi(x), \varphi(y) \rangle - \langle \varphi'(x), \varphi'(y) \rangle|$ (Rahimi & Recht, 2007; Sriperumbudur & Szabó, 2015; Sutherland & Schneider, 2015), spectral error (Avron et al., 2017; Choromanski et al., 2018; Zhang et al., 2019; Erdélyi et al., 2020; Ahle et al., 2020), and the generalization error of several learning tasks such as kernel SVM and kernel ridge regression (Avron et al., 2017; Sun et al., 2018; Li et al., 2021). A more comprehensive overview of the study of RFF can be found in a recent survey (Liu et al., 2021).

We focus on analyzing RFF with respect to the *kernel distance*. Here, the kernel distance of two data points $x, y$ is defined as their (Euclidean) distance in the feature space, i.e.,

$$\mathrm{dist}_\varphi(x, y) = \|\varphi(x) - \varphi(y)\|_2.$$

While previous results on the additive error of $K(x, y)$ (Rahimi & Recht, 2007; Sriperumbudur & Szabó, 2015; Sutherland & Schneider, 2015; Avron et al., 2017) readily implies additive error guarantee of $\mathrm{dist}_\varphi(x, y)$, the *relative* error guarantee is less understood. As far as we know, Chen & Phillips (2017) is the only previous work that gives a relative error bound for kernel distance, but unfortunately, only Gaussian kernel is studied in that work, and whether or not the kernel distance for other shift-invariant kernels is preserved by RFF, is still largely open.

In spirit, this multiplicative error guarantee of RFF, if indeed exists, makes it a kernelized version of Johnson-Lindenstrauss Lemma (Johnson & Lindenstrauss, 1984) which is one of the central result in dimension reduction. This guarantee is also very useful for downstream applications, since one can combine it directly with classical geometric algorithms such as k-means++ (Arthur & Vassilvitskii, 2007), locality sensitive hashing (Indyk & Motwani, 1998) and fast geometric matching algorithms (Raghvendra & Agarwal, 2020) to obtain very efficient algorithms for kernelized $k$-means clustering, nearest neighbor search, matching and many more.

## 1.1 OUR CONTRIBUTIONS

Our main results are characterizations of the kernel functions on which RFF preserves the kernel distance with small relative error using $\mathrm{poly}\log$ target dimensions. Furthermore, we also explore how to obtain data-oblivious dimension-reduction for kernels that cannot be handled by RFF.

As mentioned, it has been shown that RFF with small dimension preserves the *additive* error of kernel distance for all shift-invariant kernels (Rahimi & Recht, 2007; Sriperumbudur & Szabó, 2015; Sutherland & Schneider, 2015). In addition, it has been shown in Chen & Phillips (2017) that RFF indeed preserves the relative error of kernel distance for Gaussian kernels (which is shift-invariant). Hence, by analogue to the additive case and as informally claimed in Chen & Phillips (2017), one might be tempted to expect that RFF also preserves the relative error for general shift-invariant kernels as well.

**Lower Bounds.** Surprisingly, we show that this is *not* the case. In particular, we show that for a wide range of kernels, including the well-known Laplacian kernels, it requires unbounded target dimension for RFF to preserve the kernel distance with constant multiplicative error. We state the result for a Laplacian kernel in the following, and the full statement of the general conditions of kernels can be found in Theorem 4.1. In fact, what we show is a quantitatively stronger result, that if the input is $(\Delta, \rho)$-*bounded*, then preserving any constant multiplicative error requires $\Omega(\mathrm{poly}(\Delta/\rho))$ target dimension. Here, a point $x \in \mathbb{R}^d$ is $(\Delta, \rho)$-bounded if $\|x\|_\infty \leq \Delta$ and $\min_{i:x_i \neq 0} |x_i| \geq \rho$, i.e., the magnitude is (upper) bounded by $\Delta$ and the resolution is (lower) bounded by $\rho$.

**Theorem 1.1** (Lower bound; see Remark 4.1). *For every $\Delta \geq \rho > 0$ and some feature mapping $\varphi : \mathbb{R}^d \to \mathcal{H}$ of a Laplacian kernel $K(x, y) = \exp(-\|x - y\|_1)$, if for every $x, y \in \mathbb{R}^d$ that are $(\Delta, \rho)$-bounded, the RFF mapping $\pi$ for $K$ with target dimension $D$ satisfies $\mathrm{dist}_\pi(x, y) \in (1 \pm \varepsilon) \cdot \mathrm{dist}_\varphi(x, y)$ with constant probability, then $D \geq \Omega(\frac{1}{\varepsilon^2} \frac{\Delta}{\rho})$. This holds even when $d = 1$.*

**Upper Bounds.** Complementing the lower bound, we show that RFF can indeed preserve the kernel distance within $1 \pm \varepsilon$ error using $\mathrm{poly}(\varepsilon^{-1} \log n)$ target dimensions with high probability,

as long as the kernel function is shift-invariant and analytic, which includes Gaussian kernels and Cauchy kernels. Our target dimension nearly matches (up to the degree of polynomial of parameters) that is achievable by the Johnson-Lindenstrauss transform (Johnson & Lindenstrauss, 1984), which is shown to be tight (Larsen & Nelson, 2017). This upper bound also greatly generalizes the result of Chen & Phillips (2017) which only works for Gaussian kernels (see Section G for a detailed comparison).

**Theorem 1.2** (Upper bound). *Let $K : \mathbb{R}^d \times \mathbb{R}^d \to \mathbb{R}$ be a kernel function which is shift-invariant and analytic at the origin, with feature mapping $\varphi : \mathbb{R}^d \to \mathcal{H}$ for some feature space $\mathcal{H}$. For every $0 < \delta \leq \varepsilon \leq 2^{-16}$, every $d, D \in \mathbb{N}$, $D \geq \max\{\Theta(\varepsilon^{-1} \log^3(1/\delta)), \Theta(\varepsilon^{-2} \log(1/\delta))\}$, if $\pi : \mathbb{R}^d \to \mathbb{R}^D$ is an RFF mapping for $K$ with target dimension $D$, then for every $x, y \in \mathbb{R}^d$,*

$$\Pr[|\operatorname{dist}_\pi(x, y) - \operatorname{dist}_\varphi(x, y)| \leq \varepsilon \cdot \operatorname{dist}_\varphi(x, y)] \geq 1 - \delta.$$

The technical core of our analysis is a moment bound for RFF, which is derived by analysis techniques such as Taylor expansion and Cauchy's integral formula for multi-variate functions. The moment bound is slightly weaker than the moment bound of Gaussian variables, and this is the primary reason that we obtain a bound weaker than that of the Johnson-Lindenstrauss transform. Finally, several additional steps are required to fit this moment bound in Bernstein's inequality, which implies the bound in Theorem 1.2.

**Improved Dimension Bound for Kernel $k$-Means.** We show that if we focus on a specific application of kernel $k$-means, then it suffices to set the target dimension $D = \operatorname{poly}(\varepsilon^{-1} \log k)$, instead of $D = \operatorname{poly}(\varepsilon^{-1} \log n)$, to preserve the kernel $k$-means clustering cost for every $k$-partition. This follows from the probabilistic guarantee of RFF in Theorem 1.2 plus a generalization of the dimension-reduction result proved in a recent paper (Makarychev et al., 2019). Here, given a data set $P \subset \mathbb{R}^d$ and a kernel function $K : \mathbb{R}^d \times \mathbb{R}^d \to \mathbb{R}$, denoting the feature mapping as $\varphi : \mathbb{R}^d \to \mathcal{H}$, the kernel $k$-means problem asks to find a $k$-partition $\mathcal{C} := \{C_1, \ldots, C_k\}$ of $P$, such that $\operatorname{cost}^\varphi(P, \mathcal{C}) = \sum_{i=1}^k \min_{c_i \in \mathcal{H}} \sum_{x \in C_i} \|\varphi(x) - c_i\|_2^2$ is minimized.

**Theorem 1.3** (Dimension reduction for clustering; see Theorem 3.1). *For kernel $k$-means problem whose kernel function $K : \mathbb{R}^d \times \mathbb{R}^d \to \mathbb{R}$ is shift-invariant and analytic at the origin, for every data set $P \subset \mathbb{R}^d$, the RFF mapping $\pi : \mathbb{R}^d \to \mathbb{R}^D$ with target dimension $D \geq O(\frac{1}{\varepsilon^2}(\log^3 \frac{k}{\delta} + \log^3 \frac{1}{\varepsilon}))$, with probability at least $1 - \delta$, preserves the clustering cost within $1 \pm \varepsilon$ error for every $k$-partition simultaneously.*

Applying RFF to speed up kernel $k$-means has also been considered in Chitta et al. (2012), but their error bound is much weaker than ours (and theirs is not a generic dimension-reduction bound). Also, similar dimension-reduction bounds (i.e., independent of $n$) for kernel $k$-means were obtained using Nyström methods (Musco & Musco, 2017; Wang et al., 2019), but their bound is $\operatorname{poly}(k)$ which is worse than our $\operatorname{poly} \log(k)$; furthermore, our RFF-based approach is unique in that it is data-oblivious, which enables great applicability in other relevant computational settings such as streaming and distributed computing.

**Going beyond RFF.** Finally, even though we have proved RFF cannot preserve the kernel distance for every shift-invariant kernels, it does not rule out the existence of other efficient data-oblivious dimension reduction methods for those kernels, particularly for Laplacian kernel which is the primary example in our lower bound. For instance, in the same paper where RFF was proposed, Rahimi and Recht (Rahimi & Recht, 2007) also considered an alternative embedding called "binning features" that can work for Laplacian kernels. Unfortunately, to achieve a relative error of $\varepsilon$, it requires a dimension that depends *linearly* on the magnitude/aspect-ratio of the dataset, which may be exponential in the input size. Follow-up works, such as (Backurs et al., 2019), also suffer similar issues.

We make the first successful attempt towards this direction, and we show that Laplacian kernels do admit an efficient data-oblivious dimension reduction. Here, we focus on the $(\Delta, \rho)$-bounded case, Here, we use a similar setting to our lower bound (Theorem 1.1) where we focus on the $(\Delta, \rho)$-bounded case.

**Theorem 1.4** (Oblivious dimension-reduction for Laplacian kernels, see Theorem F.1). *Let $K$ be a Laplacian kernel, and denote its feature mapping as $\varphi : \mathbb{R}^d \to \mathcal{H}$. For every $0 < \delta \leq \varepsilon \leq 2^{-16}$, every $D \geq \max\{\Theta(\varepsilon^{-1} \log^3(1/\delta)), \Theta(\varepsilon^{-2} \log(1/\delta))\}$, every $\Delta \geq \rho > 0$, there is a mapping*

$\pi : \mathbb{R}^d \to \mathbb{R}^D$, *such that for every $x, y \in \mathbb{R}^d$ that are $(\Delta, \rho)$-bounded, it holds that*

$$\Pr[|\operatorname{dist}_\pi(x, y) - \operatorname{dist}_\varphi(x, y)| \leq \varepsilon \cdot \operatorname{dist}_\varphi(x, y)] \geq 1 - \delta.$$

*The time for evaluating $\pi$ is $dD \cdot \operatorname{poly}(\log \frac{\Delta}{\rho}, \log \delta^{-1})$.*

Our target dimension only depends on $\log \frac{\Delta}{\rho}$ which may be interpreted as the precision of the input. Hence, as an immediate corollary, for any $n$-points dataset with precision $1/\operatorname{poly}(n)$, we have an embedding with target dimension $D = \operatorname{poly}(\varepsilon^{-1} \log n)$, where the success probability is $1 - 1/\operatorname{poly}(n)$ and the overall running time of embedding the $n$ points is $O(n \operatorname{poly}(d\varepsilon^{-1} \log n))$.

Our proof relies on the fact that every $\ell_1$ metric space can be embedded into a squared $\ell_2$ metric space isometrically. We explicitly implement an approximate version of this embedding (Kahane, 1981), and eventually reduce our problem of Laplacian kernels to Gaussian kernels. After this reduction, we use the RFF for Gaussian kernels to obtain the final mapping. However, since the embedding to squared $\ell_2$ is only of very high dimension, to implement this whole idea efficiently, we need to utilize the special structures of the embedding, combined with an application of space bounded pseudo-random generators (PRGs) (Nisan, 1992).

Even though our algorithm utilizes the special property of Laplacian kernels and eventually still partially use the RFF for Gaussian kernels, it is still of conceptual importance. It opens up the direction of exploring general methods for Johnson-Lindenstrauss style dimension reduction for shift-invariant kernels. Furthermore, the lower bound suggests that the Johnson-Lindenstrauss style dimension reduction for general shift-invariant kernels has to be *not* differentiable, which is a fundamental difference to RFF. This requirement of "not analytical" seems very counter-intuitive, but our construction of the mapping for Laplacian kernels indeed provides valuable insights on how the non-analytical mapping behaves.

**Experiments and Comparison to Other Methods.** Apart from RFF, the Nyström and the random-projection methods are alternative popular methods for kernel dimension reduction. In Section 6, we conduct experiments to compare their empirical dimension-error tradeoffs with that of our methods on a simulated dataset. Since we focus on the error, we use the "ideal" implementation of both methods that achieve the best accuracy, so they are only in favor of the two baselines – for Nyström, we use SVD on the kernel matrix, since Nyström methods can be viewed as fast and approximate low-rank approximations to the kernel matrix; for random-projection, we apply the Johnson-Lindenstrauss transform on the explicit representations of points in the feature space. We run two experiments to compare each of RFF (on a Gaussian kernel) and our new algorithm in Theorem 1.4 (on a Laplacian kernel) with the two baselines respectively. Our experiments indicate that the Nyström method is indeed incapable of preserving the kernel distance in relative error, and more interestingly, our methods perform the best among the three, even better than the Johnson-Lindenstrauss transform which is the optimal in the worst case.

### 1.2 RELATED WORK

Variants of the vanilla RFF, particularly those that use information in the input data set and/or sample random features non-uniformly, have also been considered, including leverage score sampling random Fourier features (LSS-RFF) (Rudi et al., 2018; Liu et al., 2020; Erdélyi et al., 2020; Li et al., 2021), weighted random features (Rahimi & Recht, 2008; Avron et al., 2016; Chang et al., 2017; Dao et al., 2017), and kernel alignment (Shahrampour et al., 2018; Zhen et al., 2020).

The RFF-based methods usually work for shift-invariant kernels only. For general kernels, techniques that are based on low-rank approximation of the kernel matrix, notably Nyström method (Williams & Seeger, 2000; Gittens & Mahoney, 2016; Musco & Musco, 2017; Oglic & Gärtner, 2017; Wang et al., 2019) and incomplete Cholesky factorization (Fine & Scheinberg, 2001; Bach & Jordan, 2002; Chen et al., 2021; Jia et al., 2021)) were developed. Moreover, specific sketching techniques were known for polynomial kernels (Avron et al., 2014; Woodruff & Zandieh, 2020; Ahle et al., 2020; Song et al., 2021), a basic type of kernel that is not shift-invariant.

## 2  PRELIMINARIES

**Random Fourier Features.** RFF was first introduced by Rahimi and Recht (Rahimi & Recht, 2007). It is based on the fact that, for shift-invariant kernel $K : \mathbb{R}^d \to \mathbb{R}$ such that $K(0) = 1$ (this can be assumed w.l.o.g. by normalization), function $p : \mathbb{R}^d \to \mathbb{R}$ such that $p(\omega) = \frac{1}{2\pi} \int_{\mathbb{R}^d} K(x) e^{-\mathrm{i}\langle \omega, x \rangle} \, \mathrm{d}x$, which is the Fourier transform of $K(\cdot)$, is a probability distribution (guaranteed by Bochner's theorem (Bochner, 1933; Rudin, 1991)). Then, the RFF mapping is defined as

$$\pi(x) := \sqrt{\frac{1}{D}} \begin{pmatrix} \sin\langle \omega_1, x \rangle \\ \cos\langle \omega_1, x \rangle \\ \vdots \\ \sin\langle \omega_D, x \rangle \\ \cos\langle \omega_D, x \rangle \end{pmatrix}$$

where $\omega_1, \omega_2, \ldots, \omega_D \in \mathbb{R}^d$ are i.i.d. samples from distribution with densitiy $p$.

**Theorem 2.1** (Rahimi & Recht 2007). $\mathbb{E}[\langle \pi(x), \pi(y) \rangle] = \frac{1}{D} \sum_{i=1}^{D} \mathbb{E}[\cos\langle \omega_i, x - y \rangle] = K(x - y)$.

**Fact 2.1.** *Let $\omega$ be a random variable with distribution $p$ over $\mathbb{R}^d$. Then*

$$\forall t \in \mathbb{R}, \quad \mathbb{E}[\cos(t\langle \omega, x - y \rangle)] = \Re \int_{\mathbb{R}^d} p(\omega) e^{\mathrm{i}\langle \omega, t(x-y) \rangle} \, \mathrm{d}\omega = K(t(x - y)), \tag{1}$$

*and* $\mathrm{Var}(\cos\langle \omega, x - y \rangle) = \frac{1 + K(2(x-y)) - 2K(x-y)^2}{2}$.

## 3  UPPER BOUNDS

We present two results in this section. We start with Section 3.1 to show RFF preserves the relative error of kernel distance using $\mathrm{poly}(\varepsilon^{-1} \log n)$ target dimensions with high probability, when the kernel function is shift-invariant and analytic at origin. Then in Section 3.2, combining this bound with a generalized analysis from a recent paper (Makarychev et al., 2019), we show that RFF also preserves the clustering cost for kernel $k$-clustering problems with $\ell_p$-objective, with target dimension only $\mathrm{poly}(\varepsilon^{-1} \log k)$ which is independent of $n$.

### 3.1  PROOF OF THEOREM 1.2: THE RELATIVE ERROR FOR PRESERVING KERNEL DISTANCE

Since $K$ is shift-invariant, we interpret $K$ as a function on $\mathbb{R}^d$ instead of $\mathbb{R}^d \times \mathbb{R}^d$, such that $K(x, y) = K(x - y)$. As in Section 2, let $p : \mathbb{R}^d \to \mathbb{R}$ be the Fourier transform of $K$, and suppose in the RFF mapping $\pi$, the random variables $\omega_1, \ldots, \omega_d \in \mathbb{R}^d$ are i.i.d. sampled from the distribution with density $p$. When we say $K$ is analytic at the origin, we mean there exists some constant $r$ s.t. $K$ is analytic in $\{x \in \mathbb{R}^d : \|x\|_1 < r\}$. We pick $r_K$ to be the maximum of such constant $r$. Also notice that in $D \geq \max\{\Theta(\varepsilon^{-1} \log^3(1/\delta)), \Theta(\varepsilon^{-2} \log(1/\delta))\}$, there are constants about $K$ hidden inside the $\Theta$, i.e. $R_K$ as in Lemma 3.2.

**Fact 3.1.** *The following holds.*

- $\mathrm{dist}_\pi(x, y) = \sqrt{2 - 2/D \sum_{i=1}^{D} \cos\langle \omega_i, x - y \rangle}$, *and* $\mathrm{dist}_\varphi(x, y) = \sqrt{2 - 2K(x - y)}$.
- $\Pr[|\mathrm{dist}_\pi(x, y) - \mathrm{dist}_\varphi(x, y)| \leq \varepsilon \cdot \mathrm{dist}_\varphi(x, y)] \geq \Pr[|\mathrm{dist}_\pi(x, y)^2 - \mathrm{dist}_\varphi(x, y)^2| \leq \varepsilon \cdot \mathrm{dist}_\varphi(x, y)^2]$.

Define $X_i(x) := \cos\langle \omega_i, x \rangle - K(x)$. As a crucial step, we next analyze the moment of random variables $X_i(x - y)$. This bound will be plugged into Bernstein's inequality to conclude the proof.

**Lemma 3.1.** *If for some $r > 0$, $K$ is analytic in $\{x \in \mathbb{R}^d : \|x\|_1 < r\}$, then for every $k \geq 1$ being even and every $x$ s.t. $\|x\|_1 < r$, we have* $\mathbb{E}[|X_i(x)|^k] \leq \left( \frac{4k\|x\|_1}{r} \right)^{2k}$.

*Proof.* The proof can be found in Section A. □

**Lemma 3.2.** *For kernel $K$ which is shift-invariant and analytic at the origin, there exist $c_K, R_K > 0$ such that for all $\|x\|_1 \leq R_K$, $\frac{1-K(x)}{\|x\|_1^2} \geq \frac{c_K}{2}$.*

*Proof.* The proof can be found in Section B. $\qquad\square$

*Proof sketch of Theorem 1.2.* We present a proof sketch for Theorem 1.2, and the full proof can be found in Section C. We focus on the case when $\|x - y\|_1 \leq R_K$ (the other case can be found in the full proof). Then by Lemma 3.2, we have $2 - 2K(x - y) \geq c\|x - y\|_1^2$. Then we have:

$$\Pr\left[\left|\frac{2}{D}\sum_{i=1}^{D} X_i(x - y)\right| \leq \varepsilon \cdot (2 - 2K(x - y))\right] \geq \Pr\left[\left|\frac{2}{D}\sum_{i=1}^{D} X_i(x - y)\right| \leq c\varepsilon \cdot \|x - y\|_1^2\right].$$

We take $r = r_K$ for simplicity of exhibition. Assume $\delta \leq \min\{\varepsilon, 2^{-16}\}$, let $k = \log(2D^2/\delta), t = 64k^2/r^2$, is even. By Markov's inequality and Lemma 3.1:

$$\Pr[|X_i(x - y)| \geq t\|x - y\|_1^2] = \Pr\left[|X_i(x - y)|^k \geq t^k\|x - y\|_1^{2k}\right] \leq \frac{(4k)^{2k}}{t^k r^{2k}} = 4^{-k} \leq \frac{\delta}{2D^2}.$$

For simplicity denote $X_i(x - y)$ by $X_i$, $\|x - y\|_1^2$ by $\ell$ and define $X_i' = \mathbb{1}_{[|X_i| \geq t\ell]} t\ell \cdot \text{sgn}(X_i) + \mathbb{1}_{[|X_i| < t\ell]} X_i$, note that $\mathbb{E}[X_i] = 0$. By some further calculations and plugging in the parameters $t, \delta, D$, we can eventually obtain $\mathbb{E}[|X_i'|] \leq \delta\ell$. Denote $\sigma'^2$ as the variance of $X_i'$, then again by Lemma 3.1 we immediately have $\sigma' \leq 64\ell/r^2$. The theorem follows by a straightforward application of Bernstein's inequality. $\qquad\square$

## 3.2 DIMENSION REDUCTION FOR KERNEL CLUSTERING

We present the formal statement for Theorem 1.3 in Theorem 3.1. In fact, we consider the more general $k$-clustering problem with $\ell_2^p$-objective defined in the following Definition 3.1, which generalizes kernel $k$-means (by setting $p = 2$).

**Definition 3.1.** Given a data set $P \subset \mathbb{R}^d$ and kernel function $K : \mathbb{R}^d \times \mathbb{R}^d \to \mathbb{R}$, denoting the feature mapping as $\varphi : \mathbb{R}^d \to \mathcal{H}$, the kernel $k$-clustering problem with $\ell_p$-objective asks for a $k$-partition $\mathcal{C} = \{C_1, C_2, ..., C_k\}$ of $P$ that minimizes the cost function: $\text{cost}_p^\varphi(P, \mathcal{C}) := \sum_{i=1}^{k} \min_{c_i \in \mathcal{H}} \sum_{x \in C_i} \|\varphi(x) - c_i\|_2^p$.

**Theorem 3.1** (Generalization of Makarychev et al. 2019, Theorem 3.6). *For kernel $k$-clustering problem with $\ell_2^p$-objective whose kernel function $K : \mathbb{R}^d \times \mathbb{R}^d \to \mathbb{R}$ is shift-invariant and analytic at the origin, for every data set $P \subset \mathbb{R}^d$, the RFF mapping $\pi : \mathbb{R}^d \to \mathbb{R}^D$ with target dimension $D = \Omega(p^2 \log^3 \frac{k}{\alpha} + p^5 \log^3 \frac{1}{\varepsilon} + p^8)/\varepsilon^2$ satisfies*

$$\Pr[\forall k\text{-partition } \mathcal{C} \text{ of } P : \text{cost}_p^\pi(P, \mathcal{C}) \in (1 \pm \varepsilon) \cdot \text{cost}_p^\varphi(P, \mathcal{C})] \geq 1 - \delta.$$

*Proof.* The proof can be found in Section D. $\qquad\square$

## 4 LOWER BOUNDS

**Theorem 4.1.** *Consider $\Delta \geq \rho > 0$, and a shift-invariant kernel function $K : \mathbb{R}^d \to \mathbb{R}$, denoting its feature mapping $\varphi : \mathbb{R}^d \to \mathcal{H}$. Then there exists $x, y \in \mathbb{R}^d$ that are $(\Delta, \rho)$-bounded, such that for every $0 < \varepsilon < 1$, the RFF mapping $\pi$ for $K$ with target dimension $D$ satisfies*

$$\Pr[|\text{dist}_\varphi(x, y) - \text{dist}_\pi(x, y)| \geq \varepsilon \cdot \text{dist}_\varphi(x, y)] \geq \frac{2}{\sqrt{2\pi}} \int_{6\varepsilon\sqrt{D/s_K^{(\Delta, \rho)}}}^{\infty} e^{-s^2/2} \, ds - O\left(D^{-\frac{1}{2}}\right) \quad (2)$$

*where $s_K^{(\Delta, \rho)} := \sup_{(\Delta, \rho)\text{-bounded } x \in \mathbb{R}^d} s_K(x)$, and $s_K(x) := \frac{1 + K(2x) - 2K(x)^2}{2(1 - K(x))^2}$.*

*Proof.* The proof can be found in Section E. $\qquad\square$

Note that the right hand side of (2) is always less than 1, since the first term $\frac{2}{\sqrt{2\pi}} \int_{6\varepsilon\sqrt{D/s_K^{(\Delta,\rho)}}}^{\infty} e^{-s^2/2} \, ds$ achieves its maximum at $\frac{D\varepsilon^2}{s_K^{(\Delta,\rho)}} = 0$, and this maximum is 1. On the other hand, we need the right hand side of (2) to be $> 0$ in order to obtain a useful lower bound, and a typical setup to achieve this is when $D = \Theta\left(s_K^{(\Delta,\rho)}\right)$.

**Intuition of $s_K$.** Observe that $s_K(x)$ measures the ratio between the variance of RFF and the (squared) expectation evaluated at $x$. The intuition of considering this comes from the central limit theorem. Indeed, when the number of samples/target dimension is sufficiently large, the error/difference behaves like a Gaussian distribution where with constant probability the error $\approx \mathrm{Var}$. Hence, this $s_K$ measures the "typical" relative error when the target dimension is sufficiently large, and an upper bound of $s_K^{(\Delta,\rho)}$ is naturally a necessary condition for the bounded relative error. The following gives a simple (sufficient) condition for kernels that do not have a bounded $s_K(x)$.

*Remark* 4.1 (Simple sufficient conditions for lower bounds). Assume the input dimension is 1, so $K : \mathbb{R} \to \mathbb{R}$, and assume $\Delta = 1$, $\rho < 1$. Then the $(\Delta,\rho)$-bounded property simply requires $\rho \leq |x| \leq 1$. We claim that, if $K$'s first derivative at 0 is non-zero, i.e., $K'(0) \neq 0$, then RFF cannot preserve relative error for such $K$. To see this, we use Taylor's expansion for $K$ at the origin, and simply use the approximation to degree one, i.e., $K(x) \approx 1 + ax$ (noting that $x \leq 1$ so this is a good approximation), where $a = K'(0)$. Then

$$s_K(x) = \frac{1 + 1 + 2ax - 2(1 + ax)^2}{2a^2 x^2} = -1 - \frac{1}{ax}.$$

So if $a = K'(0) \neq 0$, then for sufficiently small $\rho$ and $|x| \geq \rho$, $s_K(\rho) \geq \Omega(1/\rho)$. This also implies the claim in Theorem 1.1 for Laplacian kernels (even though one needs to slightly modify this analysis since strictly speaking $K'$ is not well defined at 0 for Laplacian kernels). As a sanity check, for shift-invariant kernels that are analytic at the origin (which include Gaussian kernels), it is necessary that $K'(0) = 0$.

## 5 BEYOND RFF: OBLIVIOUS EMBEDDING FOR LAPLACIAN KERNEL

In this section we provide a proof sketch for theorem 1.4. A more detailed proof is deferred to section F.

**Embedding** To handle a Laplacian kernel function $K(x,y) = e^{-\frac{\|x-y\|_1}{c}}$ with some constant $c$, we cannot directly use the RFF mapping $\phi$, since our lower bound shows that the output dimension has to be very large when $K$ is not analytical around the origin. To overcome this issue, we come up with the following idea. Notice that $L(x,y)$ relies on the $\ell_1$-distance between $x,y$. If one can embed (embedding function $f$) the data points from the original $\ell_1$ metric space to a new metric space and ensure that there is an kernel function $K'$, analytical around the origin, for the new space s.t. $K(x,y) = K'(f(x), f(y))$ for every pair of original data points $x,y$, then one can use the function composition $\phi \circ f$ to get a desired mapping.

Indeed, we find that $\ell_1$ can be embedded to $\ell_2^2$ isometrically (Kahane, 1981) in the following way. Here for simplicity of exhibition we only handle the case where input data are from $\mathbb{N}^d$, upper bounded by a natural number $N$. Notice that even though input data points are only consisted of integers, the mapping construction needs to handle fractions, as we will later consider some numbers generated from Gaussian distributions or numbers computed in the RFF mapping. So we first setup two numbers, $\Delta' = \mathrm{poly}(N, \delta^{-1})$ large enough and $\rho' = 1/\mathrm{poly}(N, \delta^{-1})$ small enough. All our following operations are working on numbers that are $(\Delta', \rho')$-bounded. For each dimension we do the following transformation. Let $\pi_1 : \mathbb{N} \to \mathbb{R}^N$ be such that for every $x \in \mathbb{N}, x \leq N$, the first $x$ entries of $\pi_1(x)$ is the number 1, while all the remaining entries are 0. Then consider all $d$ dimensions. The embedding function $\pi_1^{(d)} : \mathbb{N}^d \to \mathbb{R}^{Nd}$ be such that for every $x \in \mathbb{N}^d, x_i \leq N, \forall i \in [d]$, we have $\pi_1^{(d)}(x)$ being the concatenation of $d$ vectors $\pi_1(x_i), i \in [d]$. After embedding, consider a new kernel function $K' = e^{-\frac{\|x'-y'\|_2^2}{c}}$, where $x' = \pi_1^{(d)}(x), y' = \pi_1^{(d)}(y)$. One can see immediately that $K'(x', y') = K(x, y)$. Hence, we can apply RFF then, i.e. the mapping is $\phi \circ \pi_1^{(d)}$, which has a small output dimension. Detailed proofs can be seen in section F.1.

However, there is another issue. In our setting, if the data is $(\Delta, \rho)$ bounded, then we have to pick $N = O(\frac{\Delta}{\rho})$. The computing time has a linear factor in $N$, which is too large.

**Polynomial Time Construction**  To reduce computing time, we start from the following observation about the RFF mapping $\phi(x')$. Each output dimension is actually a function of $\langle \omega, x' \rangle$, where $\omega$ is a vector of i.i.d Gaussian random variables. For simplicity of description we only consider that $x$ has only one dimension and $x' = \pi_1(x)$. So $x'$ is just a vector consists of $x$ number of 1's starting from the left and then all the remaining entries are 0's. Notice that a summation of Gaussian random variables is still a Gaussian. So given $x$, one can generate $\langle \omega, x' \rangle$ according to the summation of Gaussians. But here comes another problem. For two data points $x, y$, we need to use the same $\omega$. So if we generate $\langle \omega, x' \rangle$ and $\langle \omega', y' \rangle$ separately, then $\omega, \omega'$ are independent.

To bypass this issue, first consider the following alternate way to generate $\langle \omega, x' \rangle$. Let $h$ be the smallest integer s.t. $N \leq 2^h$. Consider a binary tree where each node has exactly 2 children. The depth is $h$. So it has exactly $2^h$ leaf nodes in the last layer. For each node $v$, we attach a random variable $\alpha_v$ in the following way. For the root, we attach a Gaussian variable which is the summation of $2^h$ independent Gaussian variable with distribution $\omega_0$. Then we proceed layer by layer from the root to leaves. For each $u, v$ being children of a common parent $w$, assume that $\alpha_w$ is the summation of $2^l$ independent $\omega_0$ distributions. Then let $\alpha_u$ be the summation of the first $2^{l-1}$ distributions among them and $\alpha_v$ be the summation of the second $2^{l-1}$ distributions. That is $\alpha_w = \alpha_u + \alpha_v$ with $\alpha_u, \alpha_v$ being independent. Notice that conditioned on $\alpha_w = a$, then $\alpha_u$ takes the value $b$ with probability $\Pr_{\alpha_u, \alpha_v \text{ i.i.d.}}[\alpha_u = b \mid \alpha_u + \alpha_v = a]$. $\alpha_v$ takes the value $a - b$ when $\alpha_u$ takes value $b$.

The randomness for generating every random variable corresponding to a node, is presented as a sequence, in the order from root to leaves, layer by layer, from left to right. We define $\alpha^x$ to be the summation of the random variables corresponding to the first $x$ leaves. Notice that $\alpha^x$ can be sampled efficiently in the following way. Consider the path from the root to the $x$-th leaf. First we sample the root, which can be computed using the corresponding part of the randomness. We use a variable $z$ to record this sample outcome, calling $z$ an accumulator for convenience. Then we visit each node along the path. When visiting $v$, assume its parent is $w$, where $\alpha_w$ has already been sampled previously with outcome $a$. If $v$ is a left child of $w$, then we sample $\alpha_v$ conditioned on $\alpha_w = a$. Assume this sampling has outcome $b$. Then we add $-a + b$ to the current accumulator $z$. If $v$ is a right child of a node $w$, then we keep the current accumulator $z$ unchanged. After visiting all nodes in the path, $z$ is the sample outcome for $\alpha^x$. We can show that the joint distribution $\alpha^x, \alpha^y$ has basically the same distribution as $\langle \omega, \pi_1(x) \rangle, \langle \omega, \pi_1(y) \rangle$. See lemma F.2.

The advantage of this alternate construction is that given any $x$, to generate $\alpha^x$, one only needs to visit the path from the root to the $x$-th leaf, using the above generating procedure. To finally reduce the time complexity, the last issue is that the uniform random string for generating random variables here is very long. If we sweep the random tape to locate the randomness used to generate a variable corresponding to a node, then we still need a linear time of $N$. Fortunately, PRGs for space bounded computation, e.g. Nisan's PRG (Nisan, 1992), can be used here to replace the uniform randomness. Because the whole procedure for deciding whether $\|\phi \circ \pi_1(x) - \phi \circ \pi_1(y)\|_2$ approximates $\sqrt{2 - K(x, y)}$ within $(1 \pm \varepsilon)$ multiplicative error, is in poly-logarithmic space. Also the computation of such PRGs can be highly efficient, i.e. given any index of its output, one can compute that bit in time polynomial of the seed length, which is poly-logarithmic of $N$. Hence the computing time of the mapping only has a factor poly-logarithmic in $N$ instead of a factor linear in $N$.

Now we have shown our construction for the case that all input data points are from $\mathbb{N}$. One can generalize this to the case where all numbers are $(\Delta, \rho)$ bounded, by doing some simple roundings and shiftings of numbers. Then this can be further generalized to the case where the input data has $d$ dimension, by simply handling each dimension and then concatenating them together. More details of this part are deferred to section F.4.

## 6  Experiments

We evaluate the empirical relative error of our methods on a simulated dataset. Specifically, we do two experiments, one to evaluate RFF on a Gaussian kernel, and the other one to evaluate the new

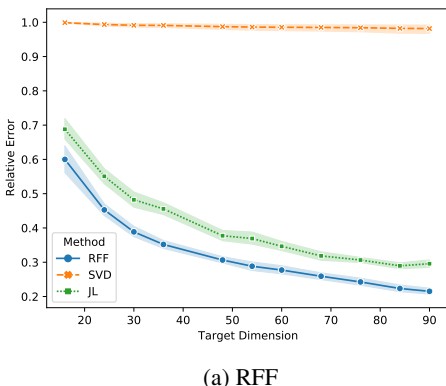

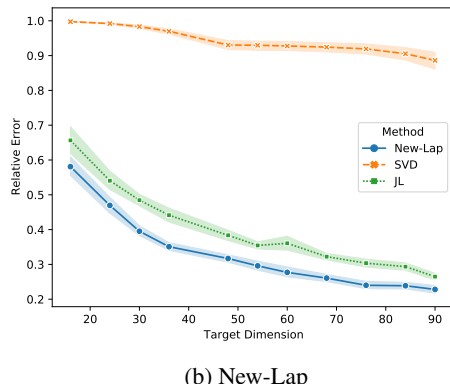

(a) RFF

(b) New-Lap

Figure 1: The dimension-error tradeoff curves for both experiments, i.e., the experiment that evaluates RFF and the one that evaluates New-Lap.

algorithm in Theorem F.1, which we call "New-Lap", on a Laplacian kernel. In each experiment, we compare against two other popular methods, particularly Nyström and random-projection methods.

**Baselines.** Observe that there are many possible implementations of these two methods. However, since we focus on the accuracy evaluation, we choose computationally-heavy but more accurate implementations as the two baselines (hence the evaluation of the error is only in the baseline's favor). In particular, we consider 1) SVD low-rank approximation which we call "SVD", and 2) the vanilla Johnson-Lindenstrauss algorithm performed on top of the high-dimensional representation of points in the feature space, which we call "JL". Note that SVD is the "ideal" goal/form of Nyström methods and that Johnson-Lindenstrauss applied on the feature space can obtain a theoretically-tight target-dimension bound (in the worst-case sense).

**Experiment Setup.** Both experiments are conducted on a synthesized dataset $X$ which consists of $N = 100$ points with $d = 60$ dimensions generated i.i.d. from a Gaussian distribution. For the experiment that we evaluate RFF, we use a Gaussian kernel $K(x) = \exp(-0.5 \cdot \|x\|^2)$, and for that we evaluate New-Lap, we use a Laplacian kernel $K(x) = \exp(-0.5 \cdot \|x\|_1)$. In each experiment, for each method, we run it for varying target dimension $D$ (for SVD, $D$ is the target rank), and we report its *empirical* relative error, which is defined as

$$\max_{x \neq y \in X} \frac{|d'(x, y) - d_K(x, y)|}{d_K(x, y)},$$

where $d_K$ is the kernel distance and $d'$ is the approximated distance. To make the result stabilized, we conduct this entire experiment for every $D$ for $T = 20$ times and report the average and 95% confident interval. We plot these dimension-error tradeoff curves, and we depict the results in Figure 1.

**Results.** We conclude that in both experiments, our methods can indeed well preserve the relative error of the kernel distance, which verifies our theorem. In particular, the dimension-error curve is comparable (and even slightly better) to the computationally heavy Johnson-Lindenstrauss algorithm (which is theoretically optimal in the worst case). On the contrary, the popular Nyström (low-rank approximation) method is largely incapable of preserving the relative error of the kernel distance. In fact, we observe that $d'_{SVD}(x, y) = 0$ or $\approx 0$ often happens for some pairs of $(x, y)$ such that $d(x, y) \neq 0$, which explains the high relative error. This indicates that our methods can indeed well preserve the kernel distance in relative error, but existing methods struggle to achieve this.

ACKNOWLEDGMENTS

Research is partially supported by a national key R&D program of China No. 2021YFA1000900, startup funds from Peking University, and the Advanced Institute of Information Technology, Peking University.

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

# Appendices

## A    PROOF OF LEMMA 3.1

**Lemma 3.1.** *If for some $r > 0$, $K$ is analytic in $\{x \in \mathbb{R}^d : \|x\|_1 < r\}$, then for every $k \geq 1$ being even and every $x$ s.t. $\|x\|_1 < r$, we have $\mathbb{E}[|X_i(x)|^k] \leq \left( \frac{4k\|x\|_1}{r} \right)^{2k}$.*

We first introduce following two lemmas to show the properties of $\mathbb{E}[X_i(x)^k]$.

**Lemma A.1.** *For any $\omega_i$ sampled in RFF and $k \geq 0$, we have $\mathbb{E}[\cos^k \langle \omega_i, x \rangle] = \frac{1}{2^k} \sum_{j=0}^{k} \binom{k}{j} K((2j - k)x)$.*

*Proof.* By eq. (1) it is sufficient to prove that

$$\cos^k \langle \omega_i, x \rangle = \frac{1}{2^k} \sum_{j=0}^{k} \binom{k}{j} \cos((2j - k)\langle \omega_i, x \rangle).$$

We prove this by induction. In the case of $k = 0$ the lemma holds obviously.

If for $k$ the lemma holds, we have

$$
\begin{aligned}
\cos^{k+1}(\langle \omega_i, x \rangle) &= \cos(\langle \omega_i, x \rangle) \cdot \frac{1}{2^k} \sum_{j=0}^{k} \binom{k}{j} \cos((2j - k)\langle \omega_i, x \rangle) \\
&= \frac{1}{2^k} \sum_{j=0}^{k} \binom{k}{j} \cos(\langle \omega_i, x \rangle) \cos(2j - k)\langle \omega_i, x \rangle) \\
&= \frac{1}{2^{k+1}} \sum_{j=0}^{k} \binom{k}{j} \Big( \cos((2j - k + 1)\langle \omega_i, x \rangle) + \cos((2j - k - 1)\langle \omega_i, x \rangle) \Big) \\
&= \frac{1}{2^{k+1}} \sum_{j=0}^{k} \binom{k}{j} \Big( \cos((2(j+1) - (k+1))\langle \omega_i, x \rangle) + \cos(2j - (k+1))\langle \omega_i, x \rangle) \Big) \\
&= \frac{1}{2^{k+1}} \sum_{j=0}^{k+1} \left( \binom{k}{j} + \binom{k}{j-1} \right) \cos((2j - (k+1))\langle \omega_i, x \rangle) \\
&= \frac{1}{2^{k+1}} \sum_{j=0}^{k+1} \binom{k+1}{j} \cos((2j - (k+1))\langle \omega_i, x \rangle)
\end{aligned}
$$

where in the third equality we use the fact that $2 \cos \alpha \cos \beta = \cos(\alpha + \beta) + \cos(\alpha - \beta)$. $\qquad \square$

**Lemma A.2.** *If there exists $r > 0$ such that $K$ is analytic in $\{x \in \mathbb{R}^d : \|x\|_1 < r\}$, then $\forall k \geq 0$, $\lim_{x \to 0} \frac{\mathbb{E}[X_i(x)^k]}{\|x\|_1^{2k}} = c$ for some constant $c$.*

*Proof.* We denote analytic function $K(x)$ as Taylor series around origin as

$$
K(x) = \sum_{\beta \in \mathbb{N}^d} c_\beta x^\beta, \tag{3}
$$

where $x^\beta := \prod_{i=1}^{d} x_i^{\beta_i}$ is a monomial and its coefficient is $c_\beta$. By definition, $c_0 = 1$ since $K(0) = 1$. We let $s : \mathbb{N}^d \to \mathbb{N}, s(\beta) := \sum_{i=1}^{d} \beta_i$ denote the degree of $x^\beta$. Since $K(x - y) = K(x, y) = K(y, x) = K(y - x)$ by definition, hence $K(x)$ is an even function, so $c_\beta = 0$ for $s(\beta)$ odd.

Recall that $X_i(x) := \cos\langle \omega_i, x \rangle - K(x)$. In the following, we drop the subscripts in $X_i, \omega_i$ and write $X, \omega$ for simplicity. By the definition of $X$ we have

$$
\mathbb{E}[X(x)^k] = \sum_{i=0}^{k} \binom{k}{i} K(x)^i (-1)^{k-i} \mathbb{E}[\cos^{k-i}\langle \omega, x \rangle]. \tag{4}
$$

Note that by Lemma A.1, $\mathbb{E}[\cos^{k-i}\langle \omega, x \rangle] = \frac{1}{2^{k-i}} \sum_{j=0}^{k-i} \binom{k-i}{j} K((2j - (k - i))x)$. Plug this in eq. (4):

$$
\begin{aligned}
\mathbb{E}[X(x)^k] &= \sum_{i=0}^{k} \binom{k}{i} K(x)^i (-1)^{k-i} \frac{1}{2^{k-i}} \sum_{j=0}^{k-i} \binom{k-i}{j} K((2j - (k - i))x) \\
&= \sum_{i=0}^{k} \binom{k}{i} K(x)^i \left( \frac{-1}{2} \right)^{k-i} \sum_{j=0}^{k-i} \binom{k-i}{j} \sum_{\beta \in \mathbb{N}^d} c_\beta (2j - k + i)^{s(\beta)} x^\beta \\
&= \sum_{\beta \in \mathbb{N}^d} c_\beta x^\beta \sum_{i=0}^{k} \binom{k}{i} K(x)^i \left( \frac{-1}{2} \right)^{k-i} \sum_{j=0}^{k-i} \binom{k-i}{j} (2j - k + i)^{s(\beta)}
\end{aligned}
$$

where the second equality comes from the Tyler expansion of $K((2j - k + i)x)$. Next we will show that $\mathbb{E}[X(x)^k]$ is of degree at least $2k$.

For $\beta = 0$ note that $\sum_{i=0}^{k} \binom{k}{i} K(x)^i (-1)^{k-i} = (K(x) - 1)^k$, since $K(x)$ is even and $K(0) = 1$, we have $\lim_{x \to 0} \frac{(K(x)-1)^k}{x^t} = 0, \forall t < 2k$ . For $\beta \neq 0$, we next show that every term of degree less than $2k$ has coefficient zero.

Fix $\beta \neq 0$ and take Tyler expansion for $K(x)^i$

$$K(x)^i = \sum_{\beta_1, \beta_2, \ldots, \beta_i} c_{\beta_1} c_{\beta_2} \ldots c_{\beta_i} x^{\beta_1 + \ldots + \beta_i},$$

Without loss of generality, we assume $\beta_{l+1}, \ldots, \beta_i$ are all $\beta$s that equals 0, so we have $c_{\beta_{l+1}} = \ldots = c_{\beta_i} = 1$.

Now we consider the coefficient of term $c_\beta c_{\beta_1} c_{\beta_2} \ldots, c_{\beta_l} x^{\beta + \sum_{j=1}^{l} \beta_j}$, which would be:

$$\tilde{C} \sum_{i=0}^{k} \binom{k}{i} \binom{i}{l} \left(\frac{-1}{2}\right)^{k-i} \sum_{j=0}^{k-i} \binom{k-i}{j} (2j - k + i)^{s(\beta)}$$

where $\tilde{C}$ is the number of ordered sequence $(\beta_1, \beta_2, \ldots, \beta_l)$, here, for $\beta_1 = \beta_2$, $(\beta_1, \beta_2, \ldots, \beta_l)$ and $(\beta_2, \beta_1, \ldots, \beta_l)$ are equivalent.

Next, we show if the degree of a monomial $s(\beta) + \sum_{j=1}^{l} s(\beta_j) < 2k$, its coefficient is zero. Since all $\beta_j \neq 0$, we may assume $s(\beta_j) \geq 2$, therefore $s(\beta) < 2k - 2l$.

Suppose operator $J$ is a mapping from a function space to itself, such that $\forall f : \mathbb{R} \to \mathbb{R}, J(f) : \mathbb{R} \to \mathbb{R}$ is defined by $J(f)(x) := f(x+1)$ . Denote $J^1 = J, J^k := J \circ J^{k-1}$ as its $k$-time composition, define $J^0$ to be the identity mapping such that $J^0(f) = f$. Similarly we can define addition that $(J_1 + J_2)(f) = J_1(f) + J_2(f)$ and scalar multiplication that $(\alpha J)(f) = \alpha(J(f))$. By definition, $cJ^m \circ J^n = J^m \circ (cJ^n) = cJ^{m+n}, \forall c \in \mathbb{R}, m, n \in \mathbb{N}$.

Let $L(x) = x^{s(\beta)}$, the coefficient can be rewritten as:

$$\tilde{C} \sum_{i=0}^{k} \binom{k}{i} \binom{i}{l} \left(\frac{-1}{2}\right)^{k-i} \sum_{j=0}^{k-i} \binom{k-i}{j} \left(J^{2j+i}(L)(-k)\right)$$

Let $P = \tilde{C} \sum_{i=0}^{k} \binom{k}{i} \binom{i}{l} \left(\frac{-1}{2}\right)^{k-i} \sum_{j=0}^{k-i} \binom{k-i}{j} J^{2j+i}(L)$, the above is $P(-k)$. Now we show $P \equiv 0$

$$
\begin{aligned}
P =& \tilde{C} \left(\sum_{i=0}^{k} \binom{k}{i} \binom{i}{l} \left(\frac{-1}{2}\right)^{k-i} J^i \circ \left(\sum_{j=0}^{k-i} \binom{k-i}{j} J^{2j}\right)\right)(L) \\
=& \tilde{C} \left(\sum_{i=l}^{k} \binom{k}{l} \binom{k-l}{i-l} J^i \circ \left(-\frac{J^0 + J^2}{2}\right)^{k-i}\right)(L) \\
=& \tilde{C} J^l \circ \binom{k}{l} \left(\sum_{i=l}^{k} \binom{k-l}{i-l} J^{i-l} \circ \left(-\frac{J^0 + J^2}{2}\right)^{k-i}\right)(L) \\
=& \tilde{C} J^l \circ \binom{k}{l} \left(J - \frac{J^0 + J^2}{2}\right)^{k-l}(L).
\end{aligned}
$$

Note that $\left(J - \frac{J^0 + J^2}{2}\right)(f)(x) = (f(x+1) - f(x))/2 - (f(x+2) - f(x+1))/2$ calculates second order difference, namely, $\forall f$ that is a polynomial of degree $k \geq 2$, $\left(J - \frac{J^0 + J^2}{2}\right)(f)$ is a polynomial of degree $k - 2$, and $\forall f$ that is a polynomial of degree $k < 2$, $\left(J - \frac{J^0 + J^2}{2}\right)(f)$ is 0.

Since $L$ is a polynomial of degree less than $2(k - l)$, we have

$$\left(J - \frac{J^0 + J^2}{2}\right)^{k-l}(L) \equiv 0.$$

Combining the above two cases, we have proved eq. (4) is of degree at least $2k$, which completes our proof. $\qquad \square$

*Proof of Lemma 3.1.* If $2k\|x\|_1 \geq r$, since $|X_i(x)| = |\cos\langle\omega_i, x\rangle - K(x)| \leq 2$, we have $\mathbb{E}[X_i(x)^k] \leq 2^k \leq \left(\frac{4k\|x\|_1}{r}\right)^{2k}$. Otherwise $\|x\|_1 < r/2k$. Define $g_k(x) := \mathbb{E}[X_i(x)^k]$, we have:

$$g_k(x) = \sum_{i=0}^{\infty} \left(\sum_{j=1}^{d} x_j \frac{\partial}{\partial x_j}\right)^i \frac{g_k(x)}{i!} = \left(\sum_{j=1}^{d} x_j \frac{\partial}{\partial x_j}\right)^{2k} \frac{g_k(\theta x)}{(2k)!}, \quad \theta \in [0,1]$$

where the second equation comes from Lemma A.2 and Taylor expansion with Lagrange remainder.

**Lemma A.3** (Cauchy's integral formula for multivariate functions Hormander 1966)**.** *For $f(z_1, ..., z_d)$ analytic in $\Delta(z, r) = \left\{\zeta = (\zeta_1, \zeta_2, \ldots, \zeta_d) \in \mathbb{C}^d; |\zeta_\nu - z_\nu| \leq r_\nu, \nu = 1, \ldots, d\right\}$*

$$f(z_1, \ldots, z_d) = \frac{1}{(2\pi i)^d} \int_{\partial D_1 \times \partial D_2 \times \cdots \times \partial D_d} \frac{f(\zeta_1, \ldots, \zeta_d)}{(\zeta_1 - z_1) \cdots (\zeta_d - z_d)} \, d\zeta.$$

*Furthermore,*

$$\frac{\partial^{k_1 + \cdots + k_d} f(z_1, z_2, \ldots, z_d)}{\partial z_1^{k_1} \cdots \partial z_d^{k_d}} = \frac{k_1! \cdots k_d!}{(2\pi i)^d} \int_{\partial D_1 \times \partial D_2 \cdots \times \partial D_d} \frac{f(\zeta_1, \ldots, \zeta_d)}{(\zeta_1 - z_1)^{k_1+1} \cdots (\zeta_d - z_d)^{k_d+1}} \, d\zeta.$$

*If in addition $|f| < M$, we have the following evaluation:*

$$\left| \frac{\partial^{k_1 + \cdots + k_d} f(z_1, z_2, \ldots, z_d)}{\partial z_1^{k_1} \cdots \partial z_d^{k_d}} \right| \leq \frac{Mk_1! \cdots k_d!}{r_1^{k_1} \cdots r_d^{k_d}}.$$

Recall that $g_k(x) = \mathbb{E}[X_i(x)^k] = \sum_{i=0}^{k} \binom{k}{i} K(x)^i (-1)^{k-i} \frac{1}{2^{k-i}} \sum_{j=0}^{k-i} \binom{k-i}{j} K((2j - (k-i))x)$, so $g_k(x) = \text{poly}(K(x), K(-x), \ldots, K(kx), K(-kx))$ is analytic when $\|x\|_1 \leq r/k$. Applying Cauchy's integral formula Lemma A.3 (here $\|z + \theta x\|_1 \leq 2 \cdot r/2k$ is in the analytic area),

$$g_k(x) = \sum_{t_1 + \cdots + t_d = 2k} \frac{x_1^{t_1} x_2^{t_2} \ldots x_d^{t_d}}{t_1! t_2! \ldots t_d!} \frac{\partial^{2k} g_k(\theta x)}{\partial x_1^{t_1} \partial x_2^{t_2} \ldots \partial x_d^{t_d}}$$

$$= \sum_{t_1 + \cdots + t_d = 2k} \frac{x_1^{t_1} x_2^{t_2} \ldots x_d^{t_d}}{(2\pi i)^d} \int_{z \in \mathbb{C}^d, |z_i| = \frac{r}{2k}} \frac{g_k(z + \theta x)}{z_1^{t_1+1} \ldots z_d^{t_d+1}} \, dz$$

we have

$$|g_k(x)| \leq \sup_{|z_i| = r/2k} |g_k(z + \theta x)| \left(\frac{2k}{r}\right)^{2k} \left|\sum_{i=1}^{d} x_i\right|^{2k} \leq \left(\frac{4k\|x\|_1}{r}\right)^{2k}.$$

$\square$

## B  PROOF OF LEMMA 3.2

**Lemma 3.2.** *For kernel $K$ which is shift-invariant and analytic at the origin, there exist $c_K, R_K > 0$ such that for all $\|x\|_1 \leq R_K$, $\frac{1 - K(x)}{\|x\|_1^2} \geq \frac{c_K}{2}$.*

*Proof.* It suffices to prove that $\liminf_{x \to 0} \frac{1 - K(x)}{\|x\|_1^2} \geq c > 0$, for some $c$. Towards proving this, we show that $K$ is *strongly* convex at origin. In fact, by definition, $\Re \int_{\mathbb{R}^d} p(\omega) e^{i\langle\omega, tx\rangle} \, d\omega = K(tx)$ for every fixed $x$, therefore $K''(tx) = \Re \int_{\mathbb{R}^d} \|\omega\|^2 p(\omega) e^{i\langle\omega, tx\rangle} \, d\omega > 0$, hence $K(tx)$ is *strongly* convex with respect to $t$ at origin, so is $K(x)$. $\square$

## C    PROOF OF THEOREM 1.2

**Theorem 1.2** (Upper bound). *Let $K : \mathbb{R}^d \times \mathbb{R}^d \to \mathbb{R}$ be a kernel function which is shift-invariant and analytic at the origin, with feature mapping $\varphi : \mathbb{R}^d \to \mathcal{H}$ for some feature space $\mathcal{H}$. For every $0 < \delta \le \varepsilon \le 2^{-16}$, every $d, D \in \mathbb{N}$, $D \ge \max\{\Theta(\varepsilon^{-1} \log^3(1/\delta)), \Theta(\varepsilon^{-2} \log(1/\delta))\}$, if $\pi : \mathbb{R}^d \to \mathbb{R}^D$ is an RFF mapping for $K$ with target dimension $D$, then for every $x, y \in \mathbb{R}^d$,*

$$\Pr[|\operatorname{dist}_\pi(x, y) - \operatorname{dist}_\varphi(x, y)| \le \varepsilon \cdot \operatorname{dist}_\varphi(x, y)] \ge 1 - \delta.$$

*Proof.* When $\|x - y\|_1 \ge R_K$, consider the function $g(t) = K(t(x-y))$. It follows from definition that $g'(0) = 0, g''(t) = -\Re \int_{\mathbb{R}^d} \|\omega\|^2 \|x - y\|^2 p(\omega) e^{i\langle \omega, t(x-y) \rangle} \, d\omega < 0$, so $g(t)$ strictly decreases for all $t > 0$. So $2 - 2K(x - y) \ge 2 - 2\max_{\|x-y\|_1 = R_K} K(x - y) > 0$. We denote $t = 2 - 2\max_{\|x-y\|_1 = R_K} K(x - y)$, so by Chernorff bound, when $D \ge \frac{1}{2t}(\ln \frac{1}{\delta} + \ln 2)$, we have:

$$\Pr\left[\left|\frac{2}{D}\sum_{i=1}^D X_i(x-y)\right| \le \varepsilon \cdot (2 - 2K(x-y))\right] \ge \Pr\left[\left|\frac{2}{D}\sum_{i=1}^D X_i(x-y)\right| \le t\right] \ge 1 - \delta.$$

When $\|x - y\|_1 \le R_K$, by Lemma 3.2, we have $2 - 2K(x - y) \ge c\|x - y\|_1^2$. Then we have:

$$\Pr\left[\left|\frac{2}{D}\sum_{i=1}^D X_i(x-y)\right| \le \varepsilon \cdot (2 - 2K(x-y))\right] \ge \Pr\left[\left|\frac{2}{D}\sum_{i=1}^D X_i(x-y)\right| \le c\varepsilon \cdot \|x - y\|_1^2\right].$$

We take $r = r_K$ for simplicity of exhibition. Assume $\delta \le \min\{\varepsilon, 2^{-16}\}$, let $k = \log(2D^2/\delta), t = 64k^2/r^2$, is even. By Markov's inequality and Lemma 3.1:

$$\Pr[|X_i(x-y)| \ge t\|x - y\|_1^2] = \Pr\left[|X_i(x-y)|^k \ge t^k\|x - y\|_1^{2k}\right] \le \frac{(4k)^{2k}}{t^k r^{2k}} = 4^{-k} \le \frac{\delta}{2D^2}.$$

For simplicity denote $X_i(x-y)$ by $X_i$, $\|x - y\|_1^2$ by $\ell$ and define $X_i' = \mathbb{1}_{[|X_i| \ge t\ell]} t\ell \cdot \operatorname{sgn}(X_i) + \mathbb{1}_{[|X_i| < t\ell]} X_i$, note that $\mathbb{E}[X_i] = 0$. Then:

$$
\begin{aligned}
|\mathbb{E}[X_i']| &\le |\mathbb{E}[X_i' \mid |X_i'| < t\ell]| \cdot \Pr[|X_i'| < t\ell] + t\ell \cdot |\Pr[X_i' \ge t\ell] - \Pr[X_i' \le -t\ell]| \\
&= |\mathbb{E}[X_i' \mid |X_i'| < t\ell]| \cdot \Pr[|X_i| < t\ell] + t\ell \cdot |\Pr[X_i \ge t\ell] - \Pr[X_i \le -t\ell]| \\
&= \frac{|\mathbb{E}[X_i] - \mathbb{E}[X_i \mid |X_i| \ge t\ell] \Pr[|X_i| \ge t\ell]|}{\Pr[|X_i| < t\ell]} \Pr[|X_i| < t\ell] + t\ell \cdot |\Pr[X_i \ge t\ell] - \Pr[X_i \le -t\ell]| \\
&= |\mathbb{E}[X_i] - \mathbb{E}[X_i \mid |X_i| \ge t\ell] \Pr[|X_i| \ge t\ell]| + t\ell \cdot |\Pr[X_i \ge t\ell] - \Pr[X_i \le -t\ell]| \\
&= |\mathbb{E}[X_i \mid |X_i| \ge t\ell] \Pr[|X_i| \ge t\ell]| + t\ell \cdot |\Pr[X_i \ge t\ell] - \Pr[X_i \le -t\ell]|
\end{aligned}
$$

where $t\ell \cdot |\Pr[X_i > t\ell] - \Pr[X_i < -t\ell]| \le t\ell \cdot \Pr[|X_i| > t\ell] \le t\ell\delta/(2D^2)$. The first inequality is by considering the two conditions $|X_i| < t\ell$ and $|X_i| \ge t\ell$, then taking a triangle inequality. The first and second equations are by definition of $X_i, X_i'$. The third equation is a straightforward computation. The last equation is due to $\mathbb{E}[X_i] = 0$. By Lemma 3.1 for every integer $\alpha$,

$$\Pr\left[|X_i| \ge \alpha\ell/r^2\right] = \Pr\left[|X_i|^{\sqrt{\alpha}/8} \ge (\alpha\ell/r^2)^{\sqrt{\alpha}/8}\right] \le \frac{\mathbb{E}[|X_i|^{\sqrt{\alpha}/8}]}{(\alpha\ell/r^2)^{\sqrt{\alpha}/8}} \le 4^{-\sqrt{\alpha}/8}.$$

The first equality is straightforward. The first inequality is by Markov. The second equality is by $\mathbb{E}[|X_i|^{\sqrt{\alpha}/8}] \le \left(\frac{\alpha\ell}{4r^2}\right)^{\sqrt{\alpha}/8}$ which follows from Lemma 3.1, and a rearrangement of parameters, where $r$ is the parameter $r$ in Lemma 3.1. Therefore,

$$
\begin{aligned}
|\mathbb{E}[X_i \mid |X_i| \ge t\ell]| \Pr[|X_i| \ge t\ell] &\le \mathbb{E}[|X_i| \mid |X_i| \ge t\ell] \Pr[|X_i| \ge t\ell] \\
&\le (t + \frac{1}{r^2})\ell \cdot \Pr[|X_i| \ge t\ell] + \frac{\ell}{r^2} \sum_{\text{integer } \alpha \ge tr^2 + 1} \Pr[|X_i| \ge \alpha\ell/r^2] \\
&\le (t + \frac{1}{r^2})\ell \cdot \frac{\delta}{2D^2} + \ell \int_{tr^2}^\infty 4^{-\sqrt{\alpha}/8} \, d\alpha \\
&\le \ell\left((t + \frac{1}{r^2})\frac{\delta}{2D^2} + \frac{16}{r^2 \ln 4} 4^{-tr^2/8}\right).
\end{aligned}
$$

The first inequality is by the property of absolute value. The second inequality is because we can divide the event $|X_i| \geq t\ell$ into $|X_i| \in [\alpha\ell/r^2, (\alpha+1)\ell/r^2), \alpha = tr^2, tr^2 + 1, \ldots$ and when $|X_i| \in [\alpha\ell/r^2, (\alpha+1)\ell/r^2), |X_i| < (\alpha+1)\ell/r^2$. The third inequality is by pluging in the previous bound for $\Pr[|X_i| \geq \alpha\ell/r^2]$. The last inequality is by a calculation of the integral.

By plugging in parameters $t, \delta, D \geq \max\{\Theta(\varepsilon^{-1}\log^3(1/\delta)), \Theta(\varepsilon^{-2}\log(1/\delta))\}$, we have $\mathbb{E}[|X_i'|] \leq \delta\ell$. Note that the $\Theta(D)$ hides a constant $r$. Denote $\sigma'^2$ as the variance of $X_i'$. So $\sigma' \leq 64\ell/r^2$ by Lemma 3.1.

**Lemma C.1** (Bernstein's Inequality). *Let $X_1, .., X_D$ be independent zero-mean random variables. Suppose that $|X_i| \leq M, \forall i$, then for all positive $t$,*

$$\Pr\left[\sum_{i=1}^{D} X_i \geq t\right] \leq \exp\left(-\frac{t^2/2}{Mt/3 + \sum_{i=1}^{D}\mathbb{E}[X_i^2]}\right).$$

Applying Bernstein's Inequality to $X_i'$,

$$\Pr\left[\sum_{i=1}^{D} X_i' - D\mathbb{E}[X_i'] \geq (c\varepsilon\ell/\sigma')D\sigma'\right] \leq \exp\left(-\frac{c^2\varepsilon^2 D}{ct\varepsilon + 2\sigma'^2/\ell^2}\right)$$

$$\leq \max\left\{\exp\left(-\frac{c\varepsilon^2 D}{2t\varepsilon}\right), \exp\left(-\frac{c^2\varepsilon^2 D}{4\sigma'^2/\ell^2}\right)\right\}.$$

Since $D \geq \max\{\Theta\left(t\varepsilon^{-1}\log(1/\delta)\right), \Theta\left(\varepsilon^{-2}\log(1/\delta)\right)\}$, we have $\Pr\left[\sum_{i=1}^{D} X_i' \geq \varepsilon(\ell/\sigma')D\sigma'\right] \leq \delta/2$. With $1 - \frac{\delta}{2}$ probability, every $X_i \leq t\ell, X_i' = X_i$. Therefore, $\Pr\left[\sum_{i=1}^{D} X_i \geq D(\delta + \varepsilon)\ell\right] \leq \delta/2$. Combine it together, $\Pr[|\operatorname{dist}_\pi(x,y) - \operatorname{dist}_\varphi(x,y)| \leq \varepsilon \cdot \operatorname{dist}_\varphi(x,y)] \geq 1 - \delta$.

$\square$

# D PROOF OF THEOREM 3.1

**Theorem 3.1** (Generalization of Makarychev et al. 2019, Theorem 3.6). *For kernel $k$-clustering problem with $\ell_2^p$-objective whose kernel function $K : \mathbb{R}^d \times \mathbb{R}^d \to \mathbb{R}$ is shift-invariant and analytic at the origin, for every data set $P \subset \mathbb{R}^d$, the RFF mapping $\pi : \mathbb{R}^d \to \mathbb{R}^D$ with target dimension $D = \Omega(p^2\log^3\frac{k}{\alpha} + p^5\log^3\frac{1}{\varepsilon} + p^8)/\varepsilon^2$ satisfies*

$$\Pr[\forall k\text{-partition } \mathcal{C} \text{ of } P : \operatorname{cost}_p^\pi(P, \mathcal{C}) \in (1 \pm \varepsilon) \cdot \operatorname{cost}_p^\varphi(P, \mathcal{C})] \geq 1 - \delta.$$

The proof relies on a key notion of $(\varepsilon, \delta, \rho)$-dimension reduction from (Makarychev et al., 2019), and we adopt it with respect to our setting/language of kernel distance as follows.

**Definition D.1** (Makarychev et al. 2019, Definition 2.1). *For $\varepsilon, \delta, \rho > 0$, a feature mapping $\varphi : \mathbb{R}^d \to \mathcal{H}$ for some Hilbert space $\mathcal{H}$, a random mapping $\pi_{d,D} : \mathbb{R}^d \to \mathbb{R}^D$ is an $(\varepsilon, \delta, \rho)$-dimension reduction, if*

- *for every $x, y \in \mathbb{R}^d$, $\frac{1}{1+\varepsilon}\operatorname{dist}_\varphi(x,y) \leq \operatorname{dist}_\pi(x,y) \leq (1+\varepsilon)\operatorname{dist}_\varphi(x,y)$ with probability at least $1 - \delta$, and*

- *for every fixed $p \in [1, \infty)$, $\mathbb{E}\left[\mathbb{1}_{\{\operatorname{dist}_\pi(x,y) > (1+\varepsilon)\operatorname{dist}_\varphi(x,y)\}}\left(\frac{\operatorname{dist}_\pi(x,y)^p}{\operatorname{dist}_\varphi(x,y)^p} - (1+\varepsilon)^p\right)\right] \leq \rho.$*

In Makarychev et al. (2019), most results are stated for a particular parameter setup of Definition D.1 resulted from Johnson-Lindenstrauss transform (Johnson & Lindenstrauss, 1984), but their analysis actually works for other similar parameter setups. The following is a generalized statement of (Makarychev et al., 2019, Theorem 3.5) which also reveals how alternative parameter setups affect the distortion. We note that this is simply a more precise and detailed statement of (Makarychev et al., 2019, Theorem 3.5), and it follows from exactly the same proof in Makarychev et al. (2019).

**Lemma D.1** (Makarychev et al. 2019, Theorem 3.5). *Let $0 < \varepsilon, \alpha < 1$ and $\theta := \min\{\varepsilon^{p+1}3^{-(p+1)(p+2)}, \alpha\varepsilon^p/(10k(1+\varepsilon)^{4p-1}), 1/10^{p+1}\}$. If some $(\varepsilon, \delta, \rho)$-dimension reduction*

$\pi$ *for feature mapping* $\varphi : \mathbb{R}^d \to \mathcal{H}$ *of some kernel function satisfies* $\delta \le \min(\theta^7/600, \theta/k)$, $\binom{k}{2}\delta \le \frac{\alpha}{2}$, $\rho \le \theta$, *then with probability at least* $1 - \alpha$, *for every partition* $\mathcal{C}$ *of* $P$,

$$\text{cost}_p^\pi(P, \mathcal{C}) \le (1 + \varepsilon)^{3p} \text{cost}_p^\varphi(P, \mathcal{C}),$$

$$(1 - \varepsilon) \text{cost}_p^\varphi(P, \mathcal{C}) \le (1 + \varepsilon)^{3p-1} \text{cost}_p^\pi(P, \mathcal{C}).$$

*Proof of Theorem 3.1.* We verify that setting $D = \Theta(\log^3 \frac{k}{\alpha} + p^3 \log^3 \frac{1}{\varepsilon} + p^6)/\varepsilon^2$, the RFF mapping $\pi$ with target dimension $D$ satisfies the conditions in Lemma D.1, namely, it is a $(\varepsilon, \delta, \rho)$-dimension reduction . In fact, Theorem 1.2 already implies such $\pi$ satisfies that for every $x, y \in \mathbb{R}^d$, $\frac{1}{1+\varepsilon} \text{dist}_\varphi(x, y) \le \text{dist}_\pi(x, y) \le (1 + \varepsilon) \text{dist}_\varphi(x, y)$ with probability at least $1 - \delta$, where $\delta = e^{-cf(\varepsilon, D)}$ for some constant $c$, and $f(\varepsilon, D) := \max\{\varepsilon^2 D, \varepsilon^{1/3} D^{1/3}\}$. For the other part,

$$\mathbb{E}\left[ \mathbb{1}_{\{\text{dist}_\pi(x,y) > (1+\varepsilon)\text{dist}_\varphi(x,y)\}} \left( \frac{\text{dist}_\pi(x,y)^p}{\text{dist}_\varphi(x,y)^p} - (1+\varepsilon)^p \right) \right]$$

$$= \int_\varepsilon^\infty \left( (1+t)^p - (1+\varepsilon)^p \right) \text{d}\left( -\Pr\left( \frac{\text{dist}_\pi(x,y)}{\text{dist}_\varphi(x,y)} > t + 1 \right) \right)$$

$$= \left[ -(1+m)^p + (1+\varepsilon)^p \right] \Pr\left( \frac{\text{dist}_\pi(x,y)}{\text{dist}_\varphi(x,y)} > m + 1 \right) \Big|_{m=\varepsilon}^{m=+\infty}$$

$$\qquad + \int_\varepsilon^\infty p(1+t)^{p-1} \Pr\left( \frac{\text{dist}_\pi(x,y)}{\text{dist}_\varphi(x,y)} > t + 1 \right) \text{d}t \qquad \text{(integration by part)}$$

$$= \int_\varepsilon^\infty p(1+t)^{p-1} \Pr\left( \frac{\text{dist}_\pi(x,y)}{\text{dist}_\varphi(x,y)} > t + 1 \right) \text{d}t$$

$$\le \int_\varepsilon^\infty p(1+t)^{p-1} e^{-cf(t,D)} \, \text{d}t.$$

Where the third equality follows by $\Pr\left( \frac{\text{dist}_\pi(x,y)}{\text{dist}_\varphi(x,y)} > m \right)$ decays exponentially fast with respect to $m$. Observe that for $p \ge 1, D \ge \frac{(p-1)^3}{8c^3}, p(1+t)^{p-1} e^{-cD^{1/3} t^{1/3}/2}$ decrease when $t \ge \varepsilon$, and for $D \ge \frac{c(p-1)}{\varepsilon^2}, p(1+t)^{p-1} e^{-ct^2 D/2}$ decrease when $t \ge \varepsilon$. Hence for $D \ge \max\{\frac{(p-1)^3}{8c^3}, \frac{c(p-1)}{\varepsilon^2}\}$, we have

$$\int_\varepsilon^\infty p(1+t)^{p-1} e^{-cf(t,D)} dt \le c' \int_\varepsilon^\infty e^{-cf(t,D)/2} dt < c'' e^{-cf(\varepsilon,D)/2}.$$

In conclusion, by setting $D = \Theta(\log^3 \frac{k}{\alpha} + p^3 \log^3 \frac{1}{\varepsilon} + p^6)/\varepsilon^2$, for $\delta = e^{-cf(\varepsilon,D)}, \rho = c'' e^{-cf(\varepsilon,D)}$ and $f(\varepsilon, D) = \max\{\varepsilon^2 D, \varepsilon^{1/3} D^{1/3}\}$, it satisfies $\delta \le \min(\theta^7/600, \theta/k), \binom{k}{2}\delta \le \frac{\alpha}{2}, \rho \le \theta$. This verifies the condition of Lemma D.1.

Finally, we conclude the proof of Theorem 3.1 by plugging $\varepsilon' = \varepsilon/3p$ and the above mentioned RFF mapping $\pi$ with target dimension $D$ into Lemma D.1. $\qquad\square$

## E   PROOF OF THEOREM 4.1

**Theorem 4.1.** *Consider* $\Delta \ge \rho > 0$, *and a shift-invariant kernel function* $K : \mathbb{R}^d \to \mathbb{R}$, *denoting its feature mapping* $\varphi : \mathbb{R}^d \to \mathcal{H}$. *Then there exists* $x, y \in \mathbb{R}^d$ *that are* $(\Delta, \rho)$-*bounded, such that for every* $0 < \varepsilon < 1$, *the RFF mapping* $\pi$ *for* $K$ *with target dimension* $D$ *satisfies*

$$\Pr[|\text{dist}_\varphi(x,y) - \text{dist}_\pi(x,y)| \ge \varepsilon \cdot \text{dist}_\varphi(x,y)] \ge \frac{2}{\sqrt{2\pi}} \int_{6\varepsilon\sqrt{D/s_K^{(\Delta,\rho)}}}^\infty e^{-s^2/2} \, \text{d}s - O\left( D^{-\frac{1}{2}} \right) \quad (2)$$

*where* $s_K^{(\Delta,\rho)} := \sup_{(\Delta, \rho)\text{-bounded } x \in \mathbb{R}^d} s_K(x)$, *and* $s_K(x) := \frac{1 + K(2x) - 2K(x)^2}{2(1 - K(x))^2}$.

*Proof.* Our proof requires the following anti-concentration inequality.

**Lemma E.1** (Berry 1941; Esseen 1942). *For i.i.d. random variables $\xi_i \in \mathbb{R}$ with mean 0 and variance 1, let $X := \frac{1}{\sqrt{D}} \sum_{i=1}^{D} \xi_i$, then for any t,*

$$\Pr[X \geq t] \geq \frac{1}{\sqrt{2\pi}} \int_t^\infty e^{-s^2/2} \, ds - O(D^{-\frac{1}{2}})$$

Let $X_i(x) := \cos\langle \omega_i, x \rangle - K(x)$, $\sigma(x) := \sqrt{\mathrm{Var}(X_i(x))} = \sqrt{\frac{1+K(2(x))-2K(x)^2}{2}}$, choose $x, y$ such that $s_K(x-y) = s_K^{(\Delta,\rho)}$. Clearly, such pair of $x, y$ satisfies that $(x-y)$ is $(\Delta, \rho)$-bounded. In fact, it is without loss of generality to assume that both $x$ and $y$ are $(\Delta, \rho)$-bounded, since one may pick $y' = 0$, $x' = x - y$ and still have $x' - y' = x - y$. We next verify that such $x, y$ satisfy our claimed properties. Indeed,

$$\Pr[|\mathrm{dist}_\varphi(x,y) - \mathrm{dist}_\pi(x,y)| \geq \varepsilon \cdot \mathrm{dist}_\varphi(x,y)]$$
$$\geq \Pr[|\mathrm{dist}_\varphi(x,y)^2 - \mathrm{dist}_\pi(x,y)^2| \geq 6\varepsilon \cdot \mathrm{dist}_\varphi(x,y)^2]$$
$$= \Pr\left[\left|\frac{2}{D} \sum_{i=1}^{D} X_i(x-y)\right| \geq 6\varepsilon(2 - 2K(x-y))\right]$$
$$= \Pr\left[\left|\frac{1}{\sqrt{D} \cdot \sigma(x-y)} \sum_{i=1}^{D} X_i(x-y)\right| \geq 6\varepsilon(1 - K(x-y)) \cdot \frac{\sqrt{D}}{\sigma(x-y)}\right]$$
$$\geq -O(D^{-1/2}) + \frac{2}{\sqrt{2\pi}} \int_{6\varepsilon(1-K(x-y))\sqrt{D}/\sigma(x-y)}^\infty e^{-s^2/2} \, ds$$
$$= -O(D^{-1/2}) + \frac{2}{\sqrt{2\pi}} \int_{6\varepsilon\sqrt{D/s_K^{(\Delta,\rho)}}}^\infty e^{-s^2/2} \, ds,$$

where the second inequality is by Lemma E.1, and the the second-last equality follows from the definition of $s_K(\cdot)$, and that of $x, y$ such that $s_K(x-y) = s_K^{(\Delta,\rho)}$. □

## F  BEYOND RFF: OBLIVIOUS EMBEDDING FOR LAPLACIAN KERNEL WITH SMALL COMPUTING TIME

In this section we show an oblivious feature mapping for Laplacian kernel dimension reduction with small computing time. The following is the main theorem.

**Theorem F.1.** *Let $K$ be a Laplacian kernel with feature mapping $\varphi : \mathbb{R}^d \to \mathcal{H}$. For every $0 < \delta \leq \varepsilon \leq 2^{-16}$, every $d, D \in \mathbb{N}$, $D \geq \max\{\Theta(\varepsilon^{-1} \log^3(1/\delta)), \Theta(\varepsilon^{-2} \log(1/\delta))\}$, every $\Delta \geq \rho > 0$, there is a mapping $\pi : \mathbb{R}^d \to \mathbb{R}^D$, such that for every $x, y \in \mathbb{R}^d$ that are $(\Delta, \rho)$-bounded,*

$$\Pr[|\mathrm{dist}_\pi(x,y) - \mathrm{dist}_\varphi(x,y)| \leq \varepsilon \cdot \mathrm{dist}_\varphi(x,y)] \geq 1 - \delta.$$

*The time for evaluating $\pi$ is $dD \, \mathrm{poly}(\log \frac{\Delta}{\rho}, \log \delta^{-1})$.*

For simplicity of exhibition, we first handle the case when the input data are from $\mathbb{N}^d$. At the end we will describe how to handle the case when the input data are from $\mathbb{R}^d$ by a simple transformation. Let $N \in \mathbb{N}$ be s.t. every entry of an input data point is at most $N$. Even though input data are only consisted of integers, the mapping construction needs to handle fractions, as we will later consider some numbers generated from Gaussian distributions or numbers computed in the RFF mapping. So we first setup two numbers, $\Delta' = \mathrm{poly}(N, \delta^{-1})$ large enough and $\rho' = 1/\mathrm{poly}(N, \delta^{-1})$ small enough. All our following operations are working on numbers that are $(\Delta', \rho')$-bounded. Denote $\rho'/\Delta'$ as $\rho_0$ for convenience.

### F.1  EMBEDDING FROM $\ell_1$ TO $\ell_2^2$

Now we describe an isometric embedding from $\ell_1$ norm to $\ell_2^2$. This construction is based on Kahane (1981), in which the first such construction of finite dimension was given, to the best of our knowledge. Let $\pi_1 : \mathbb{N} \to \mathbb{R}^N$ be such that for every $x \in \mathbb{N}$, $x \leq N$, $\pi_1(x)[j] = 1$, if $j \in [1, x]$ and

$\pi_1(x)[j] = 0$ otherwise. Let $\pi_1^{(d)} : \mathbb{N}^d \to \mathbb{R}^{Nd}$ be such that for every $x \in \mathbb{N}^d, x_i \leq N, \forall i \in [d]$, we have $\pi_1^{(d)}(x)$ being the concatenation of $d$ vectors $\pi_1(x_i), i \in [d]$.

**Lemma F.1.** *For every $x, y \in \mathbb{N}^d$ with $x_i, y_i \leq N, i \in [d]$, it holds that*

$$\|x - y\|_1 = \|\pi_1^{(d)}(x) - \pi_1^{(d)}(y)\|_2^2.$$

*Proof.* Notice that for every $i \in [d]$, $\pi_1(x_i)$ has its first $x_i$ entries being 1 while $\pi_1(y_i)$ has its first $y_i$ entries being 1. Thus $\|\pi_1(x_i) - \pi_1(y_i)\|_2^2$ is exactly $\|x_i - y_i\|_1$. If we consider all the $d$ dimensions, then by the construction of $\pi_1^{(d)}$, the lemma holds. □

## F.2 Feature Mapping for Laplacian Kernel

Notice that we can apply the mapping $\pi_1^{(d)}$ and the RFF mapping $\phi$ from Theorem 1.2 for the kernel function $\exp(-\|x-y\|_2^2)$ which is actually a Gaussian kernel. This gives a mapping which preserves kernel distance for Laplacian kernel. To be more precise, we setup the mapping to be $\pi = \phi \circ \pi_1^{(d)}$. The only drawback is that the running time is high, as in the above mapping we map $d$ dimension to $dN$ dimension. We formalize this as the following theorem.

**Theorem F.2.** *Let $K$ be a Laplacian kernel with feature map $\varphi : \mathbb{R}^d \to \mathcal{H}$. For every $0 < \delta \leq \varepsilon \leq 2^{-16}$, every $d, D, N \in \mathbb{N}$, $D \geq \max\{\Theta(\varepsilon^{-1} \log^3(1/\delta)), \Theta(\varepsilon^{-2} \log(1/\delta))\}$, there exists a mapping $\pi : \mathbb{R}^d \to \mathbb{R}^D$ s.t. for every $x, y \in \mathbb{N}^d, x, y \leq N$,*

$$\Pr[|\operatorname{dist}_\pi(x, y) - \operatorname{dist}_\varphi(x, y)| \leq \varepsilon \cdot \operatorname{dist}_\varphi(x, y)] \geq 1 - \delta.$$

*The time of evaluating $\pi$ is $\tilde{O}(dDN)$.*

*Proof.* Consider the map $\pi$ defined above. It follows by Lemma F.1 and Theorem 1.2. The running time is as stated, since we need to compute a vector of length $O(N)$ and then apply the RFF on this vector. □

The time complexity is a bit large, since we need to compute $\pi_1^{(d)}$ which has a rather large dimension $dN$. Next we show how to reduce the time complexity.

## F.3 An Alternate Construction

We give the following map $\pi'$ which has the same output distribution as that of $\pi = \phi \circ \pi_1^{(d)}$. Then in the next subsection we will use pseudorandom generators to replace the randomness in $\pi'$ while using its highly efficiency in computation to reduce the time complexity. Notice that in computing $\phi \circ \pi_1^{(d)}$, for each output dimension, the crucial step is computing $\langle \omega, \pi_1^{(d)}(x) \rangle$ for some Gaussian distribution $\omega \in \mathbb{R}^{dN}$ which has each dimension being an independent gaussian distribution $\omega_0$. The final output is a function of $\langle \omega, \pi_1^{(d)}(x) \rangle$. So we only need to present the construction for the first part, i.e. the inner product of an $N$ dimension Gaussian distribution and $\pi_1(x_1)$. For the other parts the computations are the same and finally we only need to sum them up. Hence to make the description simpler, we denote this inner product as $\langle \omega, \pi_1(x) \rangle$, where now we let $x \in \mathbb{N}, x \leq N$ and $\omega$ has $N$ dimensions each being an independent $\omega_0$.

Let $h$ be the smallest integer s.t. $N \leq 2^h$. Consider a binary tree where each node has exactly 2 children. The depth is $h$. So it has exactly $2^h \geq N$ leaf nodes in the last layer. For each node $v$, we attach a random variable $\alpha_v$ in the following way. For the root, we attach a Gaussian variable which is the summation of $2^h$ independent Gaussian variable with distribution $\omega_0$. Then we proceed layer by layer from the root to leaves. For each $u, v$ being children of a common parent $w$, assume that $\alpha_w$ is the summation of $2^l$ independent $\omega_0$ distributions. Then let $\alpha_u$ be the summation of the first $2^{l-1}$ distributions among them and $\alpha_v$ be the summation of the second $2^{l-1}$ distributions. That is $\alpha_w = \alpha_u + \alpha_v$ with $\alpha_u, \alpha_v$ being independent. Notice that conditioned on $\alpha_w = a$, then $\alpha_u$ takes the value $b$ with probability $\Pr_{\alpha_u, \alpha_v \text{ i.i.d.}}[\alpha_u = b \mid \alpha_u + \alpha_v = a]$. $\alpha_v$ takes the value $a - b$ when $\alpha_u$ takes value $b$.

The randomness for generating every random variable corresponding to a node, are presented as a sequence, in the order from root to leaves, layer by layer, from left to right. We define $\alpha^x$ to be

the summation of the random variables corresponding to the first $x$ leaves. Notice that $\alpha^x$ can be sampled efficiently in the following way. Consider the path from the root to the $x$-th leaf. First we sample the root, which can be computed using the corresponding randomness. We use a variable $z$ to record this sample outcome, calling $z$ an accumulator for convenience. Then we visit each node along the path. When visiting $v$, assume its parent is $w$, where $\alpha_w$ has already been sampled previously with outcome $a$. If $v$ is a left child of $w$, then we sample $\alpha_v$ conditioned on $\alpha_w = a$. Assume this sampling has outcome $b$. Then we add $-a + b$ to the current accumulator $z$. If $v$ is a right child of a node $w$, then we keep the current accumulator $z$ unchanged. After visiting all nodes in the path, $z$ is the sample outcome for $\alpha^x$.

**Lemma F.2.** *The joint distribution $\alpha^x, x = 0, 1, \ldots, N$ has the same distribution as $\langle \omega, \pi_1(x) \rangle, x = 0, 1, \ldots, N$.*

*Proof.* According to our construction, each leaf is an independent distribution $\omega_0$. Hence if we take all the leaves and form a vector, then it has the same distribution as $w$.

Notice that for each parent $w$ with two children $u, v$, by the construction, $\alpha_w = \alpha_u + \alpha_v$. Here $\alpha_u, \alpha_v$ are independent, each being a summation of $l$ independent $\omega_0$, with $l$ being the number of leaves derived from $u$. Thus for each layer, for every node $u$ in the layer, $\alpha_u$'s are independent and the summation of them is their parent. So for the last layer all the variables are independent and follow the distribution $\omega_0$. And for each node $w$ in the tree, $\alpha_w$ is the summation of the random variables attached to the leaves of the subtree whose root is $w$. So $\alpha^x$ is the summation of the first $x$ leaf variables. $\square$

We do the same operation for other dimensions of the output of $\pi_1^{(d)}$ and then sum them up to get an alternate construction $\pi'$ for $\pi = \phi \circ \pi_1^{(d)}$.

We note that to generate an $\alpha_v$, we only need to simulate the conditional distributions. The distribution function $F$ of the random variable is easy to derive, since its density function is a product of three Gaussian density functions, i.e.

$$\Pr_{\alpha_u, \alpha_v \text{ i.i.d.}} [\alpha_u = b \mid \alpha_u + \alpha_v = a] = \Pr_{\alpha_u, \alpha_v \text{ i.i.d.}} [\alpha_u = b, \alpha_v = a - b] / \Pr_{\alpha_u, \alpha_v \text{ i.i.d.}} [\alpha_u + \alpha_v = a]$$
$$= \Pr_{\alpha_u}[\alpha_u = b] \cdot \Pr_{\alpha_v}[\alpha_v = a - b] / \Pr_{\alpha_u, \alpha_v \text{ i.i.d.}} [\alpha_u + \alpha_v = a],$$

where $\alpha_u, \alpha_v$ are Gaussians. To compute $F$ we can use the taylor expansion of its density function to get an analytical form of $F$, and the evaluation then can be computed in time $t_\tau = \text{poly}(\rho_0^{-1})$. Recall that $\rho_0$ is defined to be $\rho'/\Delta'$. To sample $\alpha_u$, we use $\tau = O(\log \rho_0^{-1})$ uniform random bits to generate a number $p$ uniformly with precision $\text{poly}(\rho_0^{-1})$ small enough. Then we use binary search to figure out an $b$ such that $F(b) \in [p - \varepsilon_0, p + \varepsilon_0]$, for some small enough $\varepsilon_0 = \text{poly}(\rho_0)$. and the space used is $s_\tau = \text{poly} \log(\rho_0^{-1})$.

We remark that simulating a distribution using uniform random bits always has some simulating bias. The above lemma is proved under the assumption that the simulation has no bias. But we can see that the statistical distance between the simulated distribution and the original distribution is at most $\text{poly}(\rho_0) = 1/\text{poly}(N)$, which is small enough by our picking of $\Delta' = \text{poly}(N, \delta^{-1}), \rho' = 1/\text{poly}(N, \delta^{-1})$. So if we consider simulation bias, then we can show that for every subset $S \subseteq \{0, 1, \ldots, N\}$, the joint distribution $\alpha^x, x \in S$ has a statistical distance $O(|S|\varepsilon_0)$ to the joint distribution $\langle \omega, \pi_1(x) \rangle, x \in S$. Later we will only use the case that $|S| = 2$, i.e. two points. So the overall statistical distance is $\delta^{-\Theta(1)}$ which does not affect our analysis and parameters.

## F.4 Reducing the Time Complexity Using PRGs

Next we use a pseudorandom generator to replace the randomness used in the above construction. A function $G : \{0, 1\}^r \to \{0, 1\}^n$ a pseudorandom generator for space $s$ computations with error parameter $\varepsilon_g$, if for every probabilistic TM $M$ with space $s$ using $n$ bits randomness in the read-once manner

$$|\Pr[M(G(U_r)) = 1] - \Pr[M(U_n) = 1]| \le \varepsilon_g.$$

Here $r$ is called the seed length of $G$.

**Theorem F.3** (Nisan 1992). *For every $n \in \mathbb{N}$ and $s \in \mathbb{N}$, there exists an pseudorandom generator $G : \{0,1\}^r \to \{0,1\}^n$ for space $s$ computations with parameter $\varepsilon_g$, where $r = O(\log n(\log n + s + \log \frac{1}{\varepsilon_g}))$. $G$ can be computed in polynomial time (in $n, r$) and $O(r)$ space. Moreover, given an index $i \in [n]$, the $i$-th bit of the output of $G$ can be computed in time $\mathrm{poly}(r)$.*

Let $G : \{0,1\}^r \to \{0,1\}^\ell, \ell = 2dDN\tau$ be a pseudorandom generator for space $s = c_1(\log N + s_\tau)$, with $\varepsilon_g = \delta/2, \tau = c_2 \log N$ for some large enough constants $c_1, c_2$. Again we only need to consider the construction corresponding to the first output dimension of $\phi \circ \pi_1$. We replace the randomness $U_\ell$ used in the construction by output of $G$. That is, when we need $\tau$ uniform random bits to construct a distribution $\alpha_v$ in the tree, we first compute positions of these bits in $U_\ell$ and then compute the corresponding bits in the output of $G$. Then use them to do the construction in the same way. We denote this mapping using pseudorandomness as our final mapping $\pi^*$.

Now we provide a test algorithm to show that the feature mapping provided by the pseudorandom distribution has roughly the same quality as that of the mapping provided by the true randomness. We denote the test algorithm as $T = T_{K,x,y,\varepsilon}$ where $x, y \in \mathbb{R}^d$ and $K$ is a Laplacian kernel with feature mapping $\varphi$. $T$ works as the following. Its input is the randomness either being $U_\ell$ or $G(U_r)$. $T$ first computes $\mathrm{dist}_\varphi(x, y)$. Notice that $T$ actually does not have to compute $\varphi$ since the distance can be directly computed as $\sqrt{2 - 2K(x, y)}$. Then $T(G(U_r))$ computes $\mathrm{dist}_{\pi^*}(x, y)$ and test

$$|\mathrm{dist}_{\pi^*}(x, y) - \mathrm{dist}_\varphi(x, y)| \leq \varepsilon \mathrm{dist}_\varphi(x, y).$$

Notice that when the input is $U_\ell$, then this algorithm $T$ is instead testing

$$|\mathrm{dist}_{\pi'}(x, y) - \mathrm{dist}_\varphi(x, y)| \leq \varepsilon \mathrm{dist}_\varphi(x, y).$$

Recall that $\pi'$ is defined in the previous section as our mapping using true randomness.

Next we consider using $T$ on true randomness.

**Lemma F.3.** $\Pr[T(U_\ell) = 1] \geq 1 - \delta/2$.

*Proof.* By Lemma F.2, $\mathrm{dist}_{\pi'}(x, y) = \mathrm{dist}_\pi(x, y)$. By Theorem 1.2 setting the error probability to be $\delta/2$, we have

$$\Pr[|\mathrm{dist}_\pi(x, y) - \mathrm{dist}_\varphi(x, y)| \leq \varepsilon \mathrm{dist}_\varphi(x, y)] \geq 1 - \delta/2.$$

Notice that the event $T(U_\ell) = 1$ is indeed $|\mathrm{dist}_{\pi'}(x, y) - \mathrm{dist}_\varphi(x, y)| \leq \varepsilon \mathrm{dist}_\varphi(x, y)$. Hence the lemma holds. $\square$

Now we show that $T$ is actually a small space computation.

**Lemma F.4.** *$T$ runs in space $c(\log N + s_\tau)$ for some constant $c$ and the input is read-once.*

*Proof.* The computing of $\mathrm{dist}_\varphi(x, y)$ is in space $O(\log N)$, since $x, y \in \mathbb{N}, x, y \leq N$ and the kernel function $K$ can be computed in that space. Now we focus on the computation of $\pi'$. We claim that by the construction of $\alpha^x$ in section F.3, $\pi'$ can be computed using space $O(s_\tau + \log N)$. The procedure proceeds as the following. First it finds the path to the $x$-th leaf. This takes space $O(\log N)$. Then along this path, for each node we need to compute a distribution $\alpha_v$. This takes space $O(s_\tau)$. Also notice that since the randomness is presented layer by layer, the procedure only needs to do a read-once sweep of the randomness. $T$ needs to compute $\pi'$ for both $x$ and $y$, but this only blow up the space by 2. So the overall space needed is as stated. $\square$

Finally we prove our theorem by using the property of the PRG.

*Proof of Theorem F.1.* We first show our result assuming $x, y \in \mathbb{N}, x, y \leq N$ for an integer $N$. We claim that $\pi^*$ is the mapping we want. By lemma F.3, $\Pr[T(U_\ell) = 1] \geq 1 - \delta/2$. By Lemma F.4, $T$ runs in space $O(\log N + s_\tau)$ and is read-once. As $G$ is for space $c(\log N + s_\tau)$ for some large enough constant $c$,

$$|\Pr[T(G(U_r)) = 1] - \Pr[T(U_\ell) = 1]| \leq \varepsilon_g,$$

where seed length $r = O(\log(dDN s_\tau/\varepsilon_g) \log(dDN\tau))$. Notice that $T(G(U_r)) = 1$ is equivalent to $|\mathrm{dist}_{\pi^*}(x, y) - \mathrm{dist}_\varphi(x, y)| \leq \varepsilon \cdot \mathrm{dist}_\varphi(x, y)$. Thus

$$\Pr[|\mathrm{dist}_{\pi^*}(x, y) - \mathrm{dist}_\varphi(x, y)| \leq \varepsilon \cdot \mathrm{dist}_\varphi(x, y)] \geq 1 - \delta/2 - \varepsilon_g \geq 1 - \delta.$$

The running time is computed as the following. We only need to consider one dimension of the input data and one output dimension of the mapping, since others can be computed using the same time. So actually we consider the time for sampling $\alpha^x$. For $\alpha^x$, recall that we visit the path from the root to the $x$-th leaf. We don't have to compute the whole output of $G$, but instead only need to use some parts of the output. For sampling each variable $\alpha_v$ along the path, we use $\tau$ bits in the output of $G$. By Theorem F.3, the computing of each random bit in $G$'s output, given the index of this bit, needs time $\mathrm{poly}(r)$. Locating the $\tau$ bits of randomness for generating $\alpha_v$ needs time $O(\log N)$. Generating each of the Gaussian random variable using these random bits needs time $t_\tau$. Summing up these variables takes less time than sampling all of them. After sampling, the cosine and sine function of the RFF can be computed in time $\mathrm{poly}(1/\rho_0) = \mathrm{poly}(\log N, \delta^{-1})$. There are $d$ input dimensions and $D$ output dimensions. So the total time complexity is $dD\,\mathrm{poly}(\log N, \delta^{-1})$.

For the case that $x, y \in \mathbb{R}^d$, we only need to modify the embedding $\pi_1^{(d)}$ in the following way. We first round every entry so that their decimal part is now finite. The rounded parts are small enough (e.g. dropping all digits after the $10\log\rho^{-1}$-th position to the right of the decimal point.) such that this only introduce some small additive errors. Then we shift all the entries to be non-negative numbers by adding a common shift $s$. Then we multiply every entry of $x$ by a common factor $t$ s.t. every entry now only has an integer part. Notice that $t$ and $s$ can both be chosen according to $\frac{\Delta}{\rho}$, for example $t = s = O(\frac{\Delta}{\rho})$. And we can take $N$ to be $\mathrm{poly}(\frac{\Delta}{\rho})$. Then we apply $\pi_1$, and multiply a factor $\sqrt{1/t}$. Denote this map as $\tilde{\pi}_1$. Notice that this ensures that $\|x - y\|_1 = \|\tilde{\pi}_1^{(d)}(x) - \tilde{\pi}_1^{(d)}(y)\|_2^2$. Then we can apply the same construction and analysis as we did for the above natural number case. This shows the theorem. $\qquad\square$

## G    REMARKS AND COMPARISONS TO CHEN & PHILLIPS (2017)

Our upper bound in Theorem 1.2 is not directly comparable to that of Chen & Phillips (2017) which gave dimension reduction results for Gaussian kernels. Chen & Phillips (2017) showed in their Theorem 7 a slightly improved target dimension bound than ours, but it only works for the case of $\|x - y\| \geq \sigma$, where $\sigma$ is the parameter in the Gaussian kernel[1]. For the other case of $\|x - y\| < \sigma$, their Theorem 14 gave a related bound, but their guarantee is quite different from ours. Specifically, their target dimension depends linearly on the input dimension $d$. Hence, when $d$ is large (e.g., $d = \log^2 n$), this Theorem 14 is worse than ours (for the case of $\|x - y\| < \sigma$.

Finally, we remark that there might be subtle technical issues in the proof of [CP17]. Their Theorem 7 crucially uses a bound for moment generating functions that is established in their Lemma 5. However, we find various technical issues in the proof of Lemma 5 (found in their appendix). Specifically, the term $\mathbb{E}[e^{-s\frac{1}{2}\omega^2\|\Delta\|^2}]$ in the last line above "But" (in page 17), should actually be $\mathbb{E}[e^{s\frac{1}{2}\omega^2\|\Delta\|^2}]$. Even if one fixes this mistake (by negating the exponent), then eventually we can only obtain a weaker bound of $\ln M(s) \leq \frac{s^2}{4}\|\Delta\|^4 + s\|\Delta\|^2$ in the very last step, since the term $-s\|\Delta\|^2$ is negated accordingly. Hence, it is not clear if the claimed bound can still be obtained in Theorem 7.

---

[1] This condition is not clearly mentioned in the theorem statement, but it is indeed used, and is mentioned in one line above the statement in Chen & Phillips (2017).

