# OpenReview forum: "On The Relative Error of Random Fourier Features for Preserving Kernel Distance"
_ICLR.cc/2023/Conference — ICLR 2023 poster_

### Official Review · Reviewer_pJCZ · 2022-10-19

**Confidence:** 3
**Correctness:** 3
**Technical Novelty And Significance:** 3
**Empirical Novelty And Significance:** Not applicable
**Recommendation:** 8

**Clarity, Quality, Novelty And Reproducibility:**

The paper is mostly well-written in the sense that it describes the motivation and the ideas/techniques well. The overall quality looks good, though I do not have time to verify all the proofs. The main technical innovation is the upper bound in Lemma 3.1. The proof is kind of straightforward, though not trivial.

Comment on the problem background: It says that the feature map $\phi$ is from $\mathbb{R}^d$ to a general Hilbert space $H$, on the other hand the distance is defined to be the $\ell_2$ norm. Does this mean that the Hilbert space $H$ must be embeddable into $\ell_2$? It seems that the distance does not need to be $\ell_2$ norm, it just needs to be induced by the inner product on $H$.

Minor points:
- Page 2, Line -5: it would be good to elaborate on "nearly matches"
- Page 3, Line 8: “Cauchy’s evaluation formula” -> “Cauchy’s integral formula for multiple-variable functions”
Line 3 below Theorem 3.1: i.i.d. sampled -> i.i.d. samples
- Fact 3.1, last bullet point: no need to state $\operatorname{dist}_{\phi}(x,y)^2$ as this can be inferred without difficulty from the first bullet point.
- Fact 3.1: need citation or a short proof.
- Proof of Theorem 3.1, the series of inequalities in the middle of the page: the integral $\int_t^\infty 4^{-\sqrt{\alpha/8}} d\alpha$ can be calculated directly that $\int_t^\infty e^{-c\sqrt{\alpha}}d\alpha = \frac{2}{c^2}e^{-c\sqrt{t}}(1 + c\sqrt{t})$. What is written there is just a change of variable but then you still need an upper bound.
- Proof of Theorem 3.1: “By our choice of $t$ and $\delta$, … $\leq \delta l$, denote $\sigma’^2$ …, so …” -> “By our choice of $t$ and $delta$, … $\leq \delta l$. Let $\sigma’^2$ …, then …”
- Proof of Theorem 3.1, last line: “combine it together” -> “combine them together”. It would be good to be explicit about what intermediate results are combined here.
- Theorem 4.1: please rephrase the theorem statement. “For every …, every …, denoting…, there exists…” do not read very well. It might be better to say “Suppose that …”
- Page 7, line 1 after the proof of Theorem 4.1: right hand -> right-hand
- Page 7, line -7:”$\geq\Omega(1)$” -> “$ = \Omega(1)$”
- Page 7, Line -2: insert a comma after $O(1)$
- Page 8: please use different line styles - colours cannot be distinguished when printed black and white.
- Proof of Lemma 3.1: I feel there may be some small mistakes or gaps in the proof. For instance, $g_k(x)/i!$ seems to be $g_i(x)/i!$? What happens for the terms of $i < 2k$? Why do they vanish? In particular I don't see why Lemma A.2 means that they'll vanish because there is no limiting process here? Perhaps it should be an inequality here that $g_k(x)$ is dominated by the remainder term.
- Section E.1: Do NOT use $\ell_2^2$ as this typically means a two-dimensional $\ell_2$ space.


**Strength And Weaknesses:**

Strengths: Giving dimensionality reduction results for distances induced by shift-invariant kernels, where the kernel is analytic or a Laplacian kernel. The idea is simple (which is good) and some parts of the proof are nontrivial.

Weaknesses: It is not entirely convincing that the method for the Laplacian kernel is a bigger framework for other kernels. The bound $\max\{\epsilon^{-1}\log^3(1/\delta), \epsilon^{-2}\log(1/\delta)\}$ looks less satisfying.

**Summary Of The Paper:**

The paper studies dimensionality reduction techniques for estimating the kernel distance up to a (1+eps)-factor. First, the paper shows one can sample the random Fourier features to preserve $\Vert\phi(x) - \phi(y)\Vert_2$, where $\phi$ is the feature map such that $\langle \phi(x),\phi(y)\rangle = K(x,y)$ for some shift-invariant and analytic kernel $K$ and then applies to kernel k-means clustering. Then the paper shows a lower bound on the target dimension, which depends on the ‘condition number’ of the point set. The earlier upper bound requires the kernel function to be analytic at the origin, which excludes the Laplacian kernel. To remedy for this, the paper uses the fact that an $\ell_1$ metric space can be embedded into a metric space of metric $\Vert x-y\Vert_2^2$, thus turning the dimensionality reduction for a Laplacian kernel into that for a Gaussian kernel, for which the earlier random Fourier feature sampling scheme applies.

**Summary Of The Review:**

This is a solid and neat paper. I do enjoy reading it and would recommend accepting it. But I am not sure if ICLR is the correct place. Theoretical conferences may be more suitable, such as AISTATS, ICALP, ESA, RANDOM, ...

---

> ### Author Response · Authors · 2022-11-15
> **Responses to Reviewer pJCZ**
>
> We thank the reviewer for the very inspiring and insightful comments! We respond to your comments as follows.
>
>
> > It is not entirely convincing that the method for the Laplacian kernel is a bigger framework for other kernels. The bound $\max\{ \epsilon^{-1} \log^3(1 / \delta), \epsilon^{-2}\log(1 / \delta) \}$ looks less satisfying.
>
> Indeed, our oblivious dimension reduction for Laplacian kernels is more of a conceptual value, to show oblivious dimension reduction is possible for Laplacian kernels, beyond what RFF can do. This is basically to rule out the possibility that RFF is the "only way" to do oblivious dimension reduction for shift-invariant kernels, and suggests that new techniques must be developed to resolve this problem which we showcased.
>
> It is an interesting open question to find a better and more generally applicable framework for oblivious dimension reduction of general shift-invariant kernels.
>
> > Comment on the problem background: It says that the feature map $\phi$ is from $\mathbb{R}^d$ to a general Hilbert space $H$, on the other hand the distance is defined to be the $\ell_2$ norm. Does this mean that the Hilbert space $H$ must be embeddable into $\ell_2$? It seems that the distance does not need to be $\ell_2$ norm, it just needs to be induced by the inner product on $H$.
>
> Our motivation is indeed to consider the $\ell_2$ distance on the feature space, since it is widely used in downstream applications such as kernel clustering and kernel nearest-neighbor search. Hence, we define the kernel distance in a form similar to $\ell_2$.
>
> However, we emphasize that this does *not* mean we need the space to be embedded into $\ell_2$, since we can define the distance completely by the inner products, i.e., $\mathrm{dist}(x, y) := \sqrt{\langle x, x \rangle + \langle y, y \rangle - 2\langle x, y \rangle}$.
>
> Finally, we remark that for a finite dataset of $n$ points, it is always possible to find a Euclidean representation for the image of their kernel feature mapping $\varphi(x)$'s.
>
> > Page 2, Line -5: it would be good to elaborate on "nearly matches"
>
> Recall that our dimension reduction aims to preserve the pairwise $\ell_2$ distance of the data points in the feature space, within $(1 \pm \epsilon)$ error. For this task of preserving pairwise distances, the Johnson-Lindenstrauss transform, which uses $O(\epsilon^{-2}\log n)$ target dimension, is shown to be tight. Hence, comparing with this $O(\epsilon^{-2}\log n)$ target dimension bound, our bound $\mathrm{poly}(\epsilon^{-1} \log n)$ is nearly tight up to the degree of polynomial of the parameters (and certainly we do not suffer e.g. $\exp(\epsilon^{-1})$ terms).
>
> We will add this explanation in the next version.
>
>
> > Proof of Lemma 3.1: I feel there may be some small mistakes or gaps in the proof. For instance, $g_k(x) / i!$ seems to be $g_i(x) / i!$? What happens for the terms of $i < 2k$? Why do they vanish? In particular I don't see why Lemma A.2 means that they'll vanish because there is no limiting process here? Perhaps it should be an inequality here that $g_k(x)$ is dominated by the remainder term.
>
> For the term $g_k(x)/i!$, it is exactly what it is in the paper. The first equation is the Taylor expansion for $g_k(x)$.
>
> For the terms of $i<2k$, they are actually $0$ according to A.2, because since the $g_k(x)/\|x\|_1^{2k}$ is a constant when limiting $x$ to $0$, so with some effort one can show that the $i$-th derivative of $g_k(x)$ is $0$ when limiting $x$ to $0$, since this is the case for $\|x\|_1^{2k}$.
>
>
> ### Minor points
>
> > Page 3, Line 8: “Cauchy’s evaluation formula” -> ...
>
>
> > Fact 3.1, last bullet point: ...
>
> > Fact 3.1: need citation or a short proof.
>
>
> > Proof of Theorem 3.1, the series of inequalities in the middle of the page: ...
>
> > Proof of Theorem 3.1: “By our choice of $t$ and $\delta$ ... $\leq \delta l$, denote $\sigma'^2$ ..., so ..." -> "By our choice of $t$ and $\delta$, ... $\leq \delta l$. Let $\sigma'^2$ ..., then ..."
>
> > Proof of Theorem 3.1, last line: “combine it together” -> “combine them together”. It would be good to be explicit about what intermediate results are combined here.
>
> > Theorem 4.1: please rephrase the theorem statement. “For every …, every …, denoting…, there exists…” do not read very well. It might be better to say “Suppose that …”
>
> > Page 7, line 1 after the proof of Theorem 4.1: right hand -> right-hand
>
> > Page 7, line -7: "$\geq \Omega(1)$" -> "$=\Omega(1)$"
>
> > Page 7, Line -2: insert a comma after $O(1)$
>
> > Page 8: please use different line styles - colours cannot be distinguished when printed black and white.
>
>
> > Section E.1: Do NOT use $\ell_2^2$ as this typically means a two-dimensional $\ell_2$ space.
>
>
> Thanks for the very detailed comments! We will address all these in the next version.

---

### Official Review · Reviewer_vUht · 2022-10-23

**Confidence:** 4
**Correctness:** 3
**Technical Novelty And Significance:** 3
**Empirical Novelty And Significance:** 1
**Recommendation:** 8

**Clarity, Quality, Novelty And Reproducibility:**

The writing of the paper is overall clear and nice to read.

The reproducibility, for a theoretical paper, has some proofs that I'd like to see elaborated, but is overall in good condition.

I didn't review the contribution of the paper that's completely buried in the appendix.

The results are novel, though I'd like a better comparison against the prior work.

## Typos and small notes

1. Consider using a different typeface for vectors versus scalars. Not vital by any means, but I think it'd help legibility a bit.
1. [Proof of theorem 3.1, page 6, the line that starts "Assume $\delta \leq$"] I think the actual constants are mixed up here a bit. I think it's $t = 32 \log^2 (2D^2/\delta)$ and $k=\log(2D^2/\delta)$
1. [Proof of theorem 3.1, page 6, the line that starts "For simplicity denote"] Consider putting brackets in the subscript of the indicator function. I think that $1_{[|X_i| \geq t l]} tl$ reads better than $1_{|X_i| \geq t l} tl$. Also, consider using `\ell` instead of `l` in latex: $\ell$ instead of $l$.
1. [Proof of theorem 3.1, page 6, the line that starts "By Lemma 3.1 for every"] I got the rightmost probability being $(\frac 4l)^{-\sqrt{alpha}/8}$ instead of $4^{-\sqrt\alpha / 8}$. I could have messed up my algebra fwiw.
1. [Proof of theorem 3.1, page 6, the math immediately below "By Lemma 3.1 for every"] You should write out how this inequality on the right hand side appears. I assume it's some linear combination of indicator functions, but I didn't figure out how to recover this bound.
1. [Proof of theorem 3.1, page 6, the math immediately above "By out choice of $t$"] What is the rate of this integral? What happens to it? It seems to shrink in $t$, but how does that tradeoff with the $O(\frac{t\delta}{D^2})$ term next to it?
1. [Remark 4.1, bottom of page 7] At the start of the remark, $D = \Theta(...)$ has a mis-typed parenthesis

**Strength And Weaknesses:**

The paper has some really cool results in it. The first 5 pages were a really nice outline of new ideas in thinking about when and why RFF can embed pairwise distances within relative error. The first two major results, at I outlined them in the last box, are really cool and clean. Despite the issues the paper has, which I will discuss momentarily, I think those two results are strong enough to accept the paper.

There's a nice logical flow the the proofs in the paper, so my review will follow that flow.

## 1. RFF gets relative error for analytic shift-invariant kernels

The proof is a nice and intuitive statement, complete with a good sample complexity. The high level proof proceeds in three parts: use complex analysis tools to say that being analytic implies a good moment bound on each RFF feature; use being analytic to prove that the kernel has restricted strong convexity about 0; and combine these results in a fairly standard concentration argument. The burden of the proof lies in the first claim -- the moment bound -- which is proven in the appendix and I did not review in detail. Despite this, the message is pretty clean -- the proof clearly and strongly uses being analytic to create geometry that makes RFF give good accuracy near zero.

I suspect there is a very nice corollary to this relative error result, where an epsilon-net argument can give a whole subspace embedding via RFF; proving something akin to the results in [Avron et al, 2017] but with standard RFF sampling and with the regularization parameter set to zero. I'd like to know if the authors ever considered such a guarantee?

I don't know why the authors chose to prove the concentration argument underlying this result on page 6 of the paper. I checked the details, and I have a minor technical concern I'll ask later, but overall it doesn't feel very enlightening to read. Consider making a short paragraph describing the proof, pushing the details to the appendix, and recovering a bunch of space that could be used to describe the third result in the body of the paper (I'll elaborate what I mean by this later).

Either way, first result it cool and clear. All in favor.

## 2. RFF is incapable of relative error embedding for non-analytic kernels

The key proof under this claim is really simple, and just follows from the asymptotic Gaussianity of individual the RFF features (formalized via the Berry-Eissen theorem). The authors prove that all shift invariant kernels have failure probability that depends on the ratio of the standard deviation of an RFF feature to the mean value of an RFF feature, formally denoted by $s_k$ in Theorem 4.1 on page 7. When we allow two input vectors to get arbitrarily close to each other, this ratio is well controlled for analytic kernels, but may be arbitrarily bad for non-analytic kernels.

Somewhat frustratingly, in Remark 4.1 at the bottom of page 7, the authors intuitively argue why this ratio is bad for Laplacian kernels and good for analytic kernels, but do not include a formal proof. Their description makes it sound pretty easy to prove, but it really should be formalized in the appendix. A concern I have is about the hardness of _other_ non-analytic kernels. Without that proof in the appendix, it's not immediately clear to me how many other non-analytic kernels will suffer the same hardness condition. Can we state an condition on (e.g.) the Laurent coefficients of a kernel that implies RFF is incapable of giving relative error?

Either way, it's a pretty intuitive big picture, but is lacking detail in the formalization.

## 3. Pseudorandom generators and fancier ways to make JL-style guarantees for the Laplacian kernel

This result is not really described within the body of the paper. It's discussed for 5 paragraphs in the introduction ("Going beyond RFF" on page 3). But that's it. The rest is fully described in 4 pages of the appendix. I don't have spare time to review that unfortunately, so I actually have no idea how it really works or how correct it seems. 5 paragraphs of introduction ain't a lot. I'd say the proof on page 6 of the paper should be appendicized. Maybe even cut out the experiments entirely (though I realize other ICLR reviewers may not agree with this). Then the authors could fit the third contribution within the body of the paper.

I can't judge how correct it is, but it would be cool if it was correct, I guess?

## Smaller Notes and Judgements

Section 3.2 on page 7 is a short blurb about improved sample complexity for kernel clustering. The result is neat, I guess, but is not well contextualized within the world of prior work. They also call their metric an $\ell_p$-objective, where the norm really is $\|\|\cdot\|\|_2^p$, which would not usually be called an $\ell_p$ norm. Pick a new name for it, maybe call it an $\ell_2^p$-objective?

Similarly, I know that prior work on RFF like algorithms have cared about certain kinds of relative error guarantees, like in [Avron et al, 2017]. While these are cited in the paper, the results here are not well contrasted against the prior work. This is clear in Section 1.2, which is basically a block of citations without any discussion of similarity. Why does their notions of relative error subspace embedding not extend to the authors' setting? (Formally, make a subspace embedding via RFF or RFF-LS for a 2x2 kernel matrix, and see if that subspace embedding guarantee implies a relative error pairwise guarantee like what the authors are aiming for; I can clarify if this description isn't clear).

In the experiments, the authors use SVD as an algorithm to attempt to preserve pairwise distances. Why is this a benchmark? Do people really use low-rank approximation algorithms to preserve pairwise distances? This feels intuitively doomed to fail. Also, if you run an algorithm for 20 iterations, could you include measures of confidence intervals in the plot? I'm personally a fan of plotting the medians with 25th and 75th quantiles, but anything that gives a sense of typical variation would be nice.

Theorem 1.4 isn't clearly phrased. The theorem clearly creates a mapping $\pi$ which has dimension $D$ which does not depend on the precision $\frac{\Delta}{\rho}$, but the first sentence after the theorem clearly sets the dimension depends on $\log \frac\Delta\rho$.

## Summary

The paper has cool results, but is somewhat lacking in some parts of the presentation. The first results is clear and compelling. The second is compelling but a bit less clear, and the third result isn't clear at all since it's not in the body of the paper.

Nevertheless, the clear and compelling parts are absolutely strong enough to merit publication.

**Summary Of The Paper:**

The paper studies when Random Fourier Features (RFF) is able to create a Johnson-Lindenstrauss-like result for preserving the distances between the feature mappings of vectors given a shift-invariant kernel. Prior work emphasized different metrics, like getting an additive error guarantee on the pairwise distances, instead of more JL-like relative error guarantees. This paper really focuses on relative error guarantees.

Three core results are shown:
1. Suppose a kernel is both shift-invariant and analytic near zero (i.e. kernel is very smooth when it's two input are equal). Then, RFF give a relative error guarantee with a standard $O(\frac{1}{\varepsilon^2})$ sample complexity for fixed failure probability. It has a polylog dependence on failure probability where the additive error guarantees have just a log dependence, I believe.
1. Suppose a kernel is shift-invariant but not analytic near zero, like the Laplacian kernel $K(\vec x, \vec y) = e^{-\|\|\vec x - \vec y\|\|_1}$. Then, RFF for all such kernels cannot give relative error guarantees in general. If we assume the input vectors cannot be arbitrarily close to each other, then relative error guarantees become possible. Roughly speaking, two vectors being arbitrarily close to each other corresponds to approaching the non-analytic part of the shift-invariant kernel, which makes relative error estimation hard.
1. For the Laplacian kernel, a new algorithm is proposed that achieves relative error pairwise distance embedding, but not via RFF. This algorithm relies on embedding an $\ell_1$ space into a very very high dimensional $\ell_2$ space, runs RFF in some way on that $\ell_2$ space, and uses pseudo-random generators to make this computationally efficient.

**Summary Of The Review:**

## Summary

The paper has cool results, but is somewhat lacking in some parts of the presentation. The first results is clear and compelling. The second is compelling but a bit less clear, and the third result isn't clear at all since it's not in the body of the paper.

Nevertheless, the clear and compelling parts are absolutely strong enough to merit publication.

---

> ### Author Response · Authors · 2022-11-15
> **Responses to Reviewer vUht**
>
> We thank the reviewer for the very inspiring and insightful comments! We will respond to your comments following your flow of logic. Due to the word limit, we have to separate the responses into two threads.
>
> ### RFF gets relative error for analytic shift-invariant kernels
>
>
> > I suspect there is a very nice corollary to this relative error result, where an epsilon-net argument can give a whole subspace embedding via RFF; proving something akin to the results in [Avron et al, 2017] but with standard RFF sampling and with the regularization parameter set to zero. I'd like to know if the authors ever considered such a guarantee?
>
> Thanks for the suggestion! It is indeed very interesting to see if the subspace embedding can be obtained from the relative distance-error bound.
>
> However, we suspect the subspace embedding that you mention cannot be done in general. We list some evidence as follows.
> 1. It seems [Avron et al., 2017] actually gave a lower bound for the target dimension of RFF (in their Theorem 8). By a quick look, they seemed to claim that $\lambda = 0$ requires an unbounded number of dimensions for RFF.
> 2. Also, for the epsilon-net approach that you mentioned, it seems the analysis requires linearity of the embedding (where one needs to represent a point as a linear combination of net points, and apply the embedding), but the RFF unfortunately is not a linear mapping.
>
>
> > I don't know why the authors chose to prove the concentration argument underlying this result on page 6 of the paper.
>
> We agree that some of the calculations in that proof are not very interesting. In the next version, we will omit those details from the main text, change this part into a proof sketch, and try to move back the important parts of the third result into the main text.
>
> ### RFF is incapable of relative error embedding for non-analytic kernels
>
> > Somewhat frustratingly, in Remark 4.1 at the bottom of page 7, the authors intuitively argue why this ratio is bad for Laplacian kernels and good for analytic kernels, but do not include a formal proof. Their description makes it sound pretty easy to prove, but it really should be formalized in the appendix. A concern I have is about the hardness of other non-analytic kernels. Without that proof in the appendix, it's not immediately clear to me how many other non-analytic kernels will suffer the same hardness condition. Can we state an condition on (e.g.) the Laurent coefficients of a kernel that implies RFF is incapable of giving relative error?
>
> We will add a more detailed analysis of the claims made in Remark 4.1 in the next version.
>
> In fact, there are simple sufficient conditions to assert that RFF cannot preserve the relative error for certain types of kernel functions, including Laplacian kernels.
>
> For simplicity, let's assume the input dimension is $1$, so $K : \mathbb{R} \to \mathbb{R}$. Further assume $\Delta = 1$, $\rho < 1$. Then the $(\Delta, \rho)$-bounded property simply requires $\rho \leq |x| \leq 1$. We claim that, if $K$'s first derivative is non-zero, then RFF cannot preserve relative error for such $K$.
>
> To see this, we use Taylor's expansion for $K$ at the origin, and simply use the approximation to degree one, i.e., $K(x) \approx 1 + ax$ (noting that $x \leq 1$ so this is a good approximation), where $a = K'(0)$. Then
> $$
> s_K(x) = \frac{1 + 1 + 2ax - 2(1 + ax)^2}{2a^2x^2}=-1-\frac{1}{ax}
> $$
> So if $a = K'(0)\neq 0$, then for sufficiently small $\rho$ and $|x| \geq \rho$, $s_K(\rho) \geq \Omega(1 / \rho)$. This in particular implies the claim in Theorem 1.1 for Laplacian kernels.
>
> Indeed, this condition is intuitively similar to what you mentioned -- we can simply look at the coefficients of the Taylor's expansion at the origin.
>
>
> ### Pseudorandom generators and fancier ways to make JL-style guarantees for the Laplacian kernel
>
> > This result is not really described within the body of the paper.
>
> Thanks for appreciating our result! In the next version, we will follow your suggestion to shorten other not-so-important details, and try to put at least a proof sketch of the third contribution in the main text.
>
> Here's a brief overview of the high-level idea. We try to "reduce" to the Gaussian kernel case whose embedding may be obtained using RFF. While it is possible to isometrically embed $\ell_1$ to squared $\ell_2$, the target dimension could be indefinitely high, and it is unclear if this target dimension can be reduced. We bypass this issue, by applying pseudorandom generators on a carefully designed embedding to squared $\ell_2$ (which still has a high dimension but is much more "structured"), so that the final RFF on the (high-dimensional but structured) input to the Gaussian kernel may be evaluated/simulated efficiently.

---

> > ### Author Response · Authors · 2022-11-15
> > **Responses to Reviewer vUht - Continued**
> >
> > ### Other comments
> >
> > > Section 3.2 on page 7 is a short blurb about improved sample complexity for kernel clustering. The result is neat, I guess, but is not well contextualized within the world of prior work. They also call their metric an $\ell_p$-objective, where the norm really is $\|\cdot\|_2^p$, which would not usually be called an $\ell_p$-norm. Pick a new name for it, maybe call it an $\ell_2^p$-objective?
> >
> > Indeed, our terms could cause confusion. By $\ell_p$ objective we meant the aggregation function of the clustering cost is $\ell_p$, i.e., the sum $p$-th power of the distance from data points to a given center (so $p = 1$ is $k$-median, $p = 2$ is $k$-means). We will clarify this in the next version.
> >
> > > Similarly, I know that prior work on RFF like algorithms have cared about certain kinds of relative error guarantees, like in [Avron et al, 2017]. While these are cited in the paper, the results here are not well contrasted against the prior work. This is clear in Section 1.2, which is basically a block of citations without any discussion of similarity. Why does their notions of relative error subspace embedding not extend to the authors' setting? (Formally, make a subspace embedding via RFF or RFF-LS for a 2x2 kernel matrix, and see if that subspace embedding guarantee implies a relative error pairwise guarantee like what the authors are aiming for; I can clarify if this description isn't clear).
> >
> > As far as we understand, the [Avron et al., 2017] result does not really give a subspace embedding result, and it is only an approximate one. In particular, they need to suffer an "additive" $\lambda I$ in the spectral guarantee, and this $\lambda$ cannot be made $0$ in their approach as they have some dependence in the statistical dimension $s_\lambda$ as well as an $n / \lambda$ term in the target dimension. Hence, this work is related, but does not seem to be able to yield the desired relative error guarantee as we need.
> >
> > We will add this discussion in Section 1 in the next version.
> >
> > > In the experiments, the authors use SVD as an algorithm to attempt to preserve pairwise distances. Why is this a benchmark? Do people really use low-rank approximation algorithms to preserve pairwise distances? This feels intuitively doomed to fail.
> >
> > The SVD is an "ideal" form of Nystrom method, and we compare with it since Nystrom is a popular one for kernel dimension reduction. In fact, SVD/Nystrom has been shown to preserve the relative error for specific downstream applications, such as clustering [Musco and Musco, 2017]. But you are right, they are unable to preserve the relative error of pairwise kernel distance.
> >
> > > Also, if you run an algorithm for 20 iterations, could you include measures of confidence intervals in the plot?
> >
> > This is a great suggestion, and we will add the variation information to the plots in the next version.
> >
> > > Theorem 1.4 isn't clearly phrased.
> >
> > We agree, the target dimension does not depend on $\log \frac{\Delta}{\rho}$ at all, even though the running time does. We will change the order of quantification to make this clear in the next version.
> >
> > ### Typos
> >
> > > Consider using a different typeface for vectors versus scalars. Not vital by any means, but I think it'd help legibility a bit.
> >
> > > [Proof of theorem 3.1, page 6, the line that starts "Assume $\delta \leq$"] ...
> >
> > > [Proof of theorem 3.1, page 6, the line that starts "For simplicity denote"] ...
> >
> >
> > > Proof of theorem 3.1, page 6, the line that starts "By Lemma 3.1 for every"] ...
> >
> > > [Proof of theorem 3.1, page 6, the math immediately below "By Lemma 3.1 for every"] ...
> >
> > > [Proof of theorem 3.1, page 6, the math immediately above "By out choice of t"] ...
> >
> > > [Remark 4.1, bottom of page 7] At the start of the remark, $D = \Theta(\ldots)$ has a mis-typed parenthesis
> >
> >
> > Thanks for the very detailed comments, and pointing out the minor technical issues in our proof! We will do a careful pass and address all these in the next version.

---

> > ### Comment · Reviewer_vUht · 2022-11-17
> > **Thanks & follow up wrt non-analytic lower bound**
> >
> > Hi there,
> >
> > **First, a note for the meta-reviewer: this response from the authors was a delight, and I'm confidently standing by my score of 8.**
> >
> > This show of how the non-analytic lower bound flows is really nice, and totally should be in the paper as a second lemma under the main lower bound theorem statement. Maybe even just state it for kernels that have nonzero derivative? It's just very simple and compelling.
> >
> > One question comes up that feels like it should be easy-ish to see, but isn't obvious to me.
> > Why do analytic kernels have derivative 0 at 0? This seems implicit in the combination of the upper bound and the argument you present here. It seems to hold for the Gaussian kernel, but I don't see the general formulation of it.
> >
> > Thanks!

---

> > > ### Author Response · Authors · 2022-11-18
> > > **Thanks for the followup and responses to your questions**
> > >
> > > First, we thank again for your positive review and greatly appreciate your support for our paper!
> > >
> > > Yes, we agree, this simple and elegant argument should be highlighted better, and we will definitely add it around main theorem statement of the lower bound .
> > >
> > > > One question comes up that feels like it should be easy-ish to see, but isn't obvious to me. Why do analytic kernels have derivative 0 at 0? This seems implicit in the combination of the upper bound and the argument you present here. It seems to hold for the Gaussian kernel, but I don't see the general formulation of it.
> > >
> > > You are right that analytic shift-invariant kernels always have (first) derivative 0 at 0. This is guaranteed by the property of shift-invariant. Specifically, kernel function is symmetric, so K(x, y) = K(y, x), and combining with the fact that K is shift-invariant, we have K(x - y) = K(x, y) = K(y, x) = K(y - x), which implies K is an even function (so K(x) = K(-x)). This further implies K'(0) = 0. The fact that K is analytic ensures K'(0) exists.
> > >
> > > At this point, you might notice that Laplacian kernel is also shift-invariant, then why doesn't it have a zero derivative? In fact, the first derivative of Laplacian kernel does not exist (and it is not analytic). The claim K'(0) \neq 0 in my previous message is not strictly correct/well-defined, but one can replace it with K'(\delta) for \delta -> 0, and for Laplacian kernels one still has K'(\delta) = -1.

---

### Official Review · Reviewer_KzPf · 2022-10-25

**Confidence:** 4
**Correctness:** 4
**Technical Novelty And Significance:** 3
**Empirical Novelty And Significance:** 1
**Recommendation:** 6

**Clarity, Quality, Novelty And Reproducibility:**

This paper is well-written, but needs some improvements, e.g.,
- Theorem 3.1 and 1.2 (which informally states Theorem 3.1) are basically the same. There is no reason to state the same theorem twice, which can be a waste of space.
- It would be better to use the parenthesis for citing references.
- It seems that the inequality in (2) is wrong.
- It would be good to have an independent section for the "Going beyond RFF" part.

The notion of the kernel distance is not new and many technical tools are borrowed from previous works (Chen and Phillps, 2017, Makarychev et al., 2019), hence I feel that the novelty is limited.



**Strength And Weaknesses:**

Strength:

- This work extends a prior result on the Gaussian kernel to general shift-invariant kernel distances. They also study for the Laplacian kernel the bound of the number of features can be unbounded.
- The authors characterize that the kernel distance guarantee can be used for bounding the k-means clustering.
- They propose an alternative embedding for the Laplacian kernel beyond the RFF. As a result, it requires the number of features poly-log in the magnitude-resolution ratio.


Weakness:

- This paper basically improves the work of (Chen and Phillps, 2017), by generalizing the results to general shift-invariant kernels. However, importantly, the proposed result is worse than that of (Chen and Phillps, 2017), which only focuses on the Gaussian kernel. This makes the contributions weak.
- The results of the upper bound and lower bound have a different notion, i.e., the magnitude-resolution ratio only appears in the lower bound. This makes it hard to compare those bounds. Can the upper bound involve in this ratio?
- Experiments only show the relative errors on the synthetic datasets. Since the RFF has been used in many practical applications, I think more experiments should be addressed. For example, showing the k-means clustering results and comparing the Nystrom method under real-world datasets would be nice.


**Summary Of The Paper:**

This paper studies the relative error of kernel distances when applying the Random Fourier Features (RFF) to the shift-invariant kernels. While a prior work studied only the Gaussian kernel, this work generalizes the prior results to more general shift-invariant kernels. In particular, the authors show that poly(eps*log(n)) features are needed to guarantee the eps-relative errors for all the n data points. They also study a lower bound on the number of features for the Laplace kernel and show that it can be unbounded when an input has a huge magnitude-resolution ratio. In addition, they show that the RFF takes a better bound on the feature dimension on the k-means clustering compared to the Nystrom method. Finally, they provide an alternative embedding for the Laplacian kernel that requires a time in poly-log of the magnitude-resolution ratio. Experimental results evaluating the relative kernel distance under synthetic datasets are provided.


**Summary Of The Review:**

This paper extends the prior work on the Gaussian kernel to general shift-invariant ones. But, their result is worse than the prior one and technical tools are also very similar. The writing quality needs to be improved and experimental results do not well support the benefit of the RFF on the kernel distance.

---

> ### Author Response · Authors · 2022-11-15
> **Responses to Reviewer KzPf**
>
> Thank you very much for the critical and insightful comments!
>
> We start with addressing the following major concerns, followed by responses to other comments.
>
> 1. Comparison to [CP17].
> 2. Highlights of our novelty compared with the previous works.
>
> ### Comparison to [CP17]
>
> > This paper basically improves the work of (Chen and Phillps, 2017), by generalizing the results to general shift-invariant kernels. However, importantly, the proposed result is worse than that of (Chen and Phillps, 2017), which only focuses on the Gaussian kernel. This makes the contributions weak.
>
> First of all, we do not agree that [CP17] achieves a better result for the Gaussian case. As far as we understand, the result in [CP17] is not comparable to ours. In particular, the main error bound is Theorem 7 of [CP17], but it only works for $\|x - y\| \geq \sigma$, where $\sigma$ is the parameter in the Gaussian kernel. Please note that this condition is not clearly mentioned in the theorem statement, but it is indeed used and mentioned in one line above it.
>
> For the other case of $\|x - y\| < \sigma$, their Theorem 14 gave a related bound, but we find their guarantee quite different. Specifically, their target dimension depends linearly in the input dimension $d$. Hence, when $d$ is large (e.g., $d = \log^2 n$), this Theorem 14 is worse than ours (for the case of $\|x - y\| < \sigma$.
>
> In addition, we have to mention a subtle technical issue about [CP17]. Their Theorem 7 (which is the main upper bound) crucially uses a bound for momemt generating functions in Lemma 5. However, we find various technical issues in the proof of Lemma 5 (found in their appendix). Specifically, the term $\mathbb{E}[e^{-s \frac{1}{2} \omega^2 \|\Delta\|^2 }]$ in the last line above "But" (in page 17), should actually be $\mathbb{E}[e^{s \frac{1}{2} \omega^2 \|\Delta\|^2}]$. Even if one fixes this mistake (by negating the exponent), then eventually we can only obtain a weaker bound of $\ln M(s) \leq \frac{s^2}{4} \|\Delta\|^4 + s \|\Delta\|^2$ in the very last step, since the term $-s \|\Delta\|^2$ is negated accordingly. Hence, it is not clear if the claimed bound can still be obtained in Theorem 7, or at least it cannot achieve the claimed bound.
>
> Finally, we note that in our analysis we also somewhat encountered a similar technical difficulty as mentioned in the last paragraph, and to overcome it, we use a more careful analysis that details with every moment, instead of a collective upper bound of $\ln M(s)$ for the moment generating function $M(s)$. This is also a significant deviation from the techniques of [CP17].
>
> We will incorporate this discussion into our next version.
>
> ### Novelty Compared with Previous Works
>
> > The notion of the kernel distance is not new and many technical tools are borrowed from previous works (Chen and Phillps, 2017, Makarychev et al., 2019), hence I feel that the novelty is limited.
>
> As discussed in the last paragraphs, our techniques deviate from [CP17] significantly. Result-wise, our bound not only holds for general $x, y$'s (without the condition $\|x - y\| \geq \sigma$ as required by [CP17]), but also generally works for any analytic (shift-invariant) kernels (not only the Gaussian kernels as in [CP17]). Technically, these are obtained by a much more involved moment analysis, as opposed to a somewhat ad-hoc (and likely problematic) collective analysis to the moment generating function $M(s)$ in [CP17].
>
> Our clustering result is indeed heavily built on [MMR19], but it is simply an application and not considered our main result anyway.
>
> Our other results, i.e., the lower bounds and the new dimension reduction for Laplacian kernels, are both conceptually and technically new. In particular, the Laplacian kernel result turns a non-efficient/existential embedding from $\ell_1$ to squared $\ell_2$ into an efficient algorithm using pseudorandom generators on a carefully built embedding procedure, and this new algorithmic embedding from $\ell_1$ to squared $\ell_2$ may be of independent interest in the perspective of algorithm design and metric embedding.

---

> > ### Author Response · Authors · 2022-11-15
> > **Réponses to Reviewer KzPf - Continued**
> >
> > ### Other issues
> >
> > > The results of the upper bound and lower bound have a different notion, i.e., the magnitude-resolution ratio only appears in the lower bound. This makes it hard to compare those bounds. Can the upper bound involve in this ratio?
> >
> > The short answer is that the upper and lower bounds match in terms of resolution ratio, and the ratio does not appear in some of our upper bounds since they are stronger (and do not need to depend on the ratio). We highlight the difference/relation in the following.
> > 1. Theorem 1.2 gave an upper bound for *analytic* (shift-invariant) kernels, and the bound is *oblivious* to the resolution ratio, which is *stronger* than one that depends on the ratio.
> > 2. Then, the lower bound (Theorem 1.1) is mostly about non-analytic kernels which is the *complement* of Theorem 1.2, and the result is also stronger than "needed" since it says the target dimension is not only unbounded in general, but also has to be unbounded in a way that linearly depends on the resolution ratio.
> > 3. Finally, in Theorem 1.4, we show if we do not use RFF (recalling that both Theorem 1.1 and 1.2 only concern RFF), which means the lower bound (Theorem 1.1) does not apply, then the target dimension does not need to depend on the resolution ratio any more (as opposed to RFF as proved in Theorem 1.1), although our running time still depends logarithmically on it.
> >
> > > Experiments only show the relative errors on the synthetic datasets. Since the RFF has been used in many practical applications, I think more experiments should be addressed. For example, showing the k-means clustering results and comparing the Nystrom method under real-world datasets would be nice.
> >
> > For kernel $k$-means, comparisons between Nystrom/SVD and RFF have been done in previous works (e.g., [Wang, Gittens and Mahoney, JMLR 2019]), under datasets such as MNIST. The conclusion was that Nystrom/SVD performs slightly better than RFF, although not by much.
> >
> > In the next version, we will try to add an additional experiment for kernel nearest neighbor search. This is another important yet different downstream application than kernel $k$-means, since it is based more on the pairwise distance (instead of a collective guarantee as in clustering). We expect to see RFF significantly outperform other methods.
> >
> >
> >
> >
> >
> >
> > > It seems that the inequality in (2) is wrong.
> >
> > Indeed, the RHS of (2) is hard to parse, and it is not even clear that RHS is less than $1$. Hence, we provided an explanation in the paragraph below the statement of Theorem 4.1, which shows why the RHS is always at most $1$. We also discussed the typical setup of parameters such that inequality (2) is nontrivial (i.e., significantly greater than $0$).
> >
> > We hope this helps to resolve your concerns. If still not, could you please further elaborate on why you think this is wrong?
> >
> >
> > > It would be good to have an independent section for the "Going beyond RFF" part.
> >
> > We agree, and we will try to move it from the appendix to the main text in the next version.
> >
> > > Theorem 3.1 and 1.2 (which informally states Theorem 3.1) are basically the same. There is no reason to state the same theorem twice, which can be a waste of space.
> >
> > > It would be better to use the parenthesis for citing references.
> >
> > Thanks for the suggestions and we will implement these in the next version.

---

> > > ### Comment · Reviewer_vUht · 2022-11-17
> > > **My 2 Cents**
> > >
> > > Hi there, I'm not the reviewer who wrote the first comment here, but I did read these short few messages, and I want to champion this paper a bit.
> > >
> > > The response from the authors seems very well written and clear. I haven't gone back into the prior work to verify the concerns about [CP17], but if somebody asks me to, then I would.
> > >
> > > ---
> > >
> > > First, even assuming everything Reviewer KzPf said is technically correct, the first weakness they describe sounds odd to me:
> > > - _The prior work had a stronger result for only the Gaussian kernel_ doesn't weaken the results of this paper which apply to all analytic shift-invariant kernels. It provides a broader theoretical explanation for why the Gaussian kernel has this nice structure. Conceptually, we can imagine dozens of ways that the Gaussian kernel is special (e.g. it's shift invariant, log-concave, infinitely differentiable, has small tails, is Lipschitz, etc.), and this paper nails the explanation down to just *shift-invariant and analytic*. ***That's super cool.***
> > > - On top of that, the authors' rebuttal refutes the claim that the prior work actually gives such stronger results
> > >
> > > ---
> > >
> > > The reviewer's second concern, that the upper and lower bounds don't compare, doesn't quiet hold up either. They describe a nice intuition for the comparison of the bounds on page 7, and in their response to me on openreview, they present a short formal argument for it.
> > >
> > > Also, the reviewer just said the following without elaborating....
> > > > It seems that the inequality in (2) is wrong.
> > >
> > > _what's wrong?_
> > >
> > > ---
> > >
> > > The third concern is that the experiments are on synthetic data. This is more subjective, but I'm personally frustrated by the constant push to have theoretical ML papers expected to have experiments at all when the paper's goal is to explain ideas. RFF and JL and Nystrom are extremely well established algorithms, and they've been studied in practice in other works. It's not clear why we expect this paper to have these experiments, but it's a bonus when they do.

---

### Decision · Program_Chairs · 2023-01-20

**Decision:**

Accept: poster

**Justification For Why Not Higher Score:**



**Justification For Why Not Lower Score:**



**Metareview: Summary, Strengths And Weaknesses:**

Though kernel methods are among the most powerful techniques of our times with a large number of successful applications, this flexibility can be restricted by their scalability. In this submission the authors focus on the well-known and popular acceleration scheme of (continuous bounded translation-invariant) kernels called random Fourier features (RFF). Particularly, they pursue the derivation of relative error guarantees for the kernel distance, as defined in the 1st displayed equation of page 2. The authors present both negative and positive results in this direction. They show that (i) for instance for the Laplacian kernel (Theorem 1.1 and Theorem 4.1) preserving small relative error using low dimension is not feasible, (ii) but for analytic kernels it is the case (Theorem 1.2 and Theorem 3.1), (iii) the latter result can also be propagated to kernel k-means (Theorem 1.3 and Theorem 3.2), and (iv) Laplacian kernels are amenable to efficient data-oblivious dimension reduction (Theorem 1.4 and Theorem E.1).

The importance of kernel methods is inevitable; analyzing scalable kernel techniques is of paramount interest to the machine learning community. The authors present valuable new results in this direction (with focus on the theoretical side) as it was assessed by the reviewers. As it was also pointed out by the reviewers, the manuscript would benefit from slight adjustments: removal of 'close-to-repetitive' statements, and using the resulting space to one of the main results (Theorem E.1) currently hidden in the supplement.

**Note From Pc:**

if the above contains the word "oral" or "spotlight" please see: "oral" presentation means -> notable-top-5% and "spotlight" means -> notable-top-25%. As stated in our emails, we are disassociating presentation type from AC recommendations